# Potentiation of cerebellar Purkinje cells facilitates whisker reflex adaptation through increased simple spike activity

Vincenzo Romano[1†], Licia De Propris[1,2†], Laurens WJ Bosman[1†]*, Pascal Warnaar[1], Michiel M ten Brinke[1], Sander Lindeman[1], Chiheng Ju[1], Arthiha Velauthapillai[1], Jochen K Spanke[1], Emily Middendorp Guerra[1], Tycho M Hoogland[1,3], Mario Negrello[1], Egidio D'Angelo[2,4], Chris I De Zeeuw[1,3]*

[1]Department of Neuroscience, Erasmus MC, Rotterdam, The Netherlands; [2]Department of Brain and Behavioral Sciences, University of Pavia, Pavia, Italy; [3]Netherlands Institute for Neuroscience, Royal Academy of Arts and Sciences, Amsterdam, The Netherlands; [4]Brain Connectivity Center, Instituto Fondazione C Mondino, Pavia, Italy

**Abstract** Cerebellar plasticity underlies motor learning. However, how the cerebellum operates to enable learned changes in motor output is largely unknown. We developed a sensory-driven adaptation protocol for reflexive whisker protraction and recorded Purkinje cell activity from crus 1 and 2 of awake mice. Before training, simple spikes of individual Purkinje cells correlated during reflexive protraction with the whisker position without lead or lag. After training, simple spikes and whisker protractions were both enhanced with the spiking activity now leading behavioral responses. Neuronal and behavioral changes did not occur in two cell-specific mouse models with impaired long-term potentiation at their parallel fiber to Purkinje cell synapses. Consistent with cerebellar plasticity rules, increased simple spike activity was prominent in cells with low complex spike response probability. Thus, potentiation at parallel fiber to Purkinje cell synapses may contribute to reflex adaptation and enable expression of cerebellar learning through increases in simple spike activity.

DOI: https://doi.org/10.7554/eLife.38852.001

**\*For correspondence:**
l.bosman@erasmusmc.nl (LWJB);
c.dezeeuw@erasmusmc.nl (CIDZ)

[†]These authors contributed equally to this work

**Competing interests:** The authors declare that no competing interests exist.

## Introduction

Active touch is important for exploring our environment, allowing us to assess the shape, substance and movements of objects and organisms around us (*Prescott et al., 2011*). Throughout the animal kingdom, various systems have evolved for this purpose; these include for example the antennae of insects, the fingertips of primates and the well-developed whisker systems of rodents and sea mammals (*Ahl, 1986*; *Dehnhardt et al., 2001*; *Staudacher et al., 2005*; *Dere et al., 2007*; *Anjum and Brecht, 2012*). Activation of these sensory organs can provoke reactive movements, often occurring as a reflex (*Nguyen and Kleinfeld, 2005*; *Bellavance et al., 2017*; *Brown and Raman, 2018*; *Staudacher et al., 2005*). For survival, it is important to maintain optimal control of such reflexes in daily life and to be able to adapt these movements (*Voigts et al., 2015*; *Anjum and Brecht, 2012*; *Arkley et al., 2017*).

Given the impact of cerebellar plasticity on a wide variety of motor learning tasks (*Herzfeld et al., 2015*; *Herzfeld et al., 2018*; *Medina and Lisberger, 2008*; *ten Brinke et al., 2015*; *Thier et al., 2002*; *Voges et al., 2017*; *Yang and Lisberger, 2017*), it can be anticipated that adaptation of reflexive whisker movements is also partly controlled by plastic processes in the cerebellum. Historically, most studies on cerebellar learning have suggested that long-term depression

**eLife digest** Rodents use their whiskers to explore the world around them. When the whiskers touch an object, it triggers involuntary movements of the whiskers called whisker reflexes. Experiencing the same sensory stimulus multiple times enables rodents to fine-tune these reflexes, e.g., by making their movements larger or smaller. This type of learning is often referred to as motor learning.

A part of the brain called cerebellum controls motor learning. It contains some of the largest neurons in the nervous system, the Purkinje cells. Each Purkinje cell receives input from thousands of extensions of small neurons, known as parallel fibers. It is thought that decreasing the strength of the connections between parallel fibers and Purkinje cells can help mammals learn new movements. This is the case in a type of learning called Pavlovian conditioning. It takes its name from the Russian scientist, Pavlov, who showed that dogs can learn to salivate in response to a bell signaling food. Pavlovian conditioning enables animals to optimize their responses to sensory stimuli.

But Romano et al. now show that increasing the strength of connections between parallel fibers and Purkinje cells can also support learning. To trigger reflexive whisker movements, a machine blew puffs of air onto the whiskers of awake mice. After repeated exposure to the air puffs, the mice increased the size of their whisker reflexes. At the same time, their Purkinje cells became more active and the connections between Purkinje cells and parallel fibers grew stronger. Artificially increasing Purkinje cell activity triggered the same changes in whisker reflexes as the air puffs themselves.

Textbooks still report that only weakening of connections within the cerebellum enables animals to learn and modify movements. The data obtained by Romano al. thus paint a new picture of how the cerebellum works in the context of whisker learning. They show that strengthening these connections can also support movement-related learning.
DOI: https://doi.org/10.7554/eLife.38852.002

(LTD) at the parallel fiber to Purkinje cell (PC) synapse may act as the main cellular mechanism underlying induction of cerebellar motor learning (*Albus, 1971*; *Konnerth et al., 1992*; *Ito, 1989*; *Koekkoek et al., 2003*; *Medina and Lisberger, 2008*; *Boele et al., 2018*; *Narain et al., 2018*). However, parallel fiber LTD is unlikely to be the sole cellular mechanism underlying cerebellar learning (*Gao et al., 2012*; *Hansel et al., 2001*; *D'Angelo et al., 2016*). Short-term forms of plasticity probably also contribute, as some forms of behavioral adaptation can be linked to changes in PC activity during the previous trial (*Yang and Lisberger, 2014*; *Herzfeld et al., 2018*). Moreover, long-term potentiation (LTP) of parallel fiber to PC synapses may also be relevant, as various PC-specific mutants with impaired LTP show deficits in cerebellar learning (*Schonewille et al., 2010*; *Schonewille et al., 2011*; *Rahmati et al., 2014*; *Gutierrez-Castellanos et al., 2017*). Possibly, different cerebellar cellular mechanisms dominate the induction of different forms of learning, dependent on the requirements of the downstream circuitries involved (*De Zeeuw and Ten Brinke, 2015*; *Suvrathan et al., 2016*).

While many studies have focused on the synaptic mechanism(s) that may induce cerebellar motor learning, the spiking mechanisms that are responsible for the expression thereof remain relatively unexplored. To date, whereas evidence is emerging that the expression of conditioned eyeblink responses is mediated by a long-lasting *suppression* of simple spikes of PCs in the deep fissure of lobule simplex (*Heiney et al., 2014*; *Halverson et al., 2015*; *ten Brinke et al., 2015*), it is unclear to what extent enduring *increases* in simple spike activity can also contribute to the expression of cerebellar learning, and if so for what forms of learning. Here, we developed a novel whisker training paradigm that is likely to generate plasticity in the cerebellar cortex and to produce increases in simple spike activity at the PC level following induction of LTP at the parallel fiber to PC synapse (*D'Angelo et al., 2001*; *Lev-Ram et al., 2002*; *Lev-Ram et al., 2003*; *Coesmans et al., 2004*; *Ramakrishnan et al., 2016*; *van Beugen et al., 2013*). Indeed, we show that a brief period of 4 Hz air-puff stimulation of the whiskers can enhance touch-induced whisker protraction as well as PC simple spike firing for tens of minutes. Moreover, these behavioral and neuronal changes are both

absent in two independent mouse mutant lines deficient for parallel fiber to PC LTP, bridging the putative mechanism of memory expression with that of memory induction.

## Results

### Touch-induced whisker protraction

The large facial whiskers are a prime source of sensory information for many mammals, in particular for rodents that can make elaborate movements with their large facial whiskers (*Arkley et al., 2017*; *Brecht, 2007*; *Welker, 1964*; *Bosman et al., 2011*; *Vincent, 1913*). It has been noted that passive touch can trigger active whisker movements in mice (*Bellavance et al., 2017*; *Brown and Raman, 2018*; *Nguyen and Kleinfeld, 2005*; *Ferezou et al., 2007*), but this behavior has not been described in great detail yet. Here, we studied whisker movements following rostro-caudal air-puff stimulation of the whisker pad in 16 awake, head-restrained mice (*Figure 1A–C*). The air-puffer was placed in such a way that most, if not all, large mystacial whiskers were affected by the air flow from the front. The mice made active whisker protractions following the retractions induced by the air flow in the large majority (82%) of stimulus trials (*Figure 1D–E*; *Figure 1—figure supplement 1C*). Because of the systematic full-field air-flow from the front, the touch-induced protraction was typically performed by all whiskers simultaneously (*data not shown*), which is in line with the presumed reflexive nature of this movement (*Bellavance et al., 2017*; *Brown and Raman, 2018*; *Nguyen and Kleinfeld, 2005*). Moreover, as reported previously (*Ferezou et al., 2007*), the touch-induced whisker protraction was followed in about half the trials (51%) by extended periods of active whisker movements during the subsequent 200 ms interval (*Figure 1D*; *Figure 1—figure supplement 1A–C*). However, under our experimental conditions with a 2 s inter-trial interval, spontaneous whisking in between the stimuli was relatively rare. Across all 16 mice measured, we found spontaneous movements (with an amplitude exceeding 10°) only in 12% of the 100 trials per mouse during the 200 ms interval prior to stimulus onset.

To find out whether touch-induced whisker protraction can indeed be described as a reflex (*Bellavance et al., 2017*; *Brown and Raman, 2018*; *Nguyen and Kleinfeld, 2005*), we wanted to know to what extent the movements also show signs characteristic of startle responses or voluntary events, which have a different identity. A startle response would be expected to be not only highly stereotypic, but to also show relatively little direction-specificity, and to reveal signs of pre-pulse inhibition (*Gogan, 1970*; *Swerdlow et al., 1992*; *Moreno-Paublete et al., 2017*). Instead, if the air-puff triggered a conscious, explorative movement, the animal would most likely make spontaneous movements towards the source of the air-puff, dependent on its specific position. To explore these possibilities, we placed a second air-puffer at the caudal side of the whisker field and a third air-puffer at the front of the contralateral whisker field, and we provided air-puffs from the three different orientations, intermingling trials with and without brief pre-pulses in random order (*Figure 1—figure supplement 2A*). An air-puff from the front on the ipsilateral side induced a retraction prior to the active protraction. Such a retraction was mostly absent when stimulating from the back. Contralateral stimulation also evoked a slight retraction, followed by a much larger forward sweep (*Figure 1—figure supplement 2B–E*; *Supplementary file 1A*). Thus, applying the air-puff from different angles produced different retractions and different subsequent protractions, arguing against a stereotypical startle behavior that occurs independent from the stimulus conditions. Moreover, we did not observe a diminishing effect of the weaker pre-pulse on the reaction to the stronger pulse (p=0.268; Dunn's pairwise post-hoc test after Friedman's ANOVA; p=0.003; Fr = 13.933; df = 3). Finally, we also did not observe distinct explorative movements linked to the puff sources, which might have suggested dominant voluntary components (*Figure 1—figure supplement 2B–C*). Altogether, the reactive nature of the touch-induced whisker movements in the absence of characteristic signs of startle or voluntary responses indicates that the air-puff induced protraction is indeed a reflexive movement.

### Anatomical distribution of Purkinje cell responses to whisker pad stimulation

In line with the fact that PCs receive sensory whisker input not only directly from the brainstem but also indirectly from thalamo-cortical pathways (*Figure 2F*) (*Kleinfeld et al., 1999*; *McElvain et al.,*

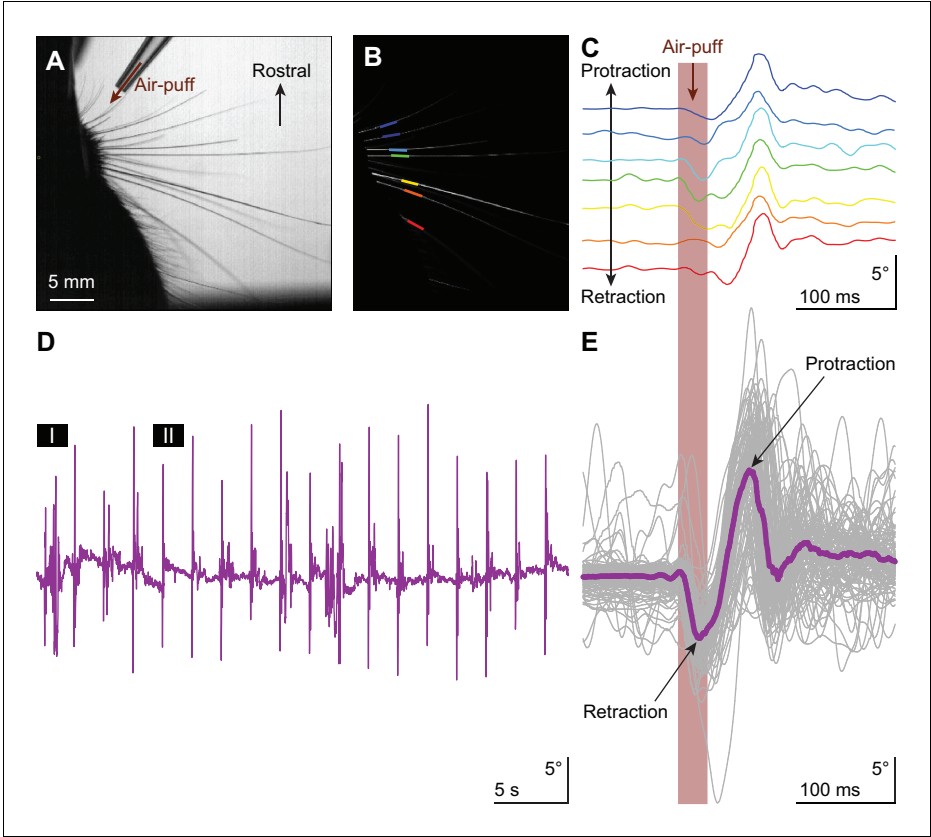

**Figure 1.** Touch-induced whisker protraction A brief (30 ms) air-puff to the whisker pad induces a reflexive protraction of all large mystacial whiskers. Our experiments were performed in awake, head-restrained mice that had all whiskers intact. (**A**) Photograph showing a part of the mouse head with the large facial whiskers and the location and direction of the air-puffer (top). (**B**) The large facial whiskers were recognized in high-speed videos (1 kHz full-frame rate) by a tracking algorithm and individual whiskers are color-coded. (**C**) Air-puff stimulation triggered stereotypic whisker movements consisting of an initial backward movement followed by active protraction. Deflection angles of individually tracked whiskers are denoted in distinct colors (same color scheme as in (**B**)). (**D**) The mean whisker angle during 0.5 Hz air-puff stimulation of the whisker pad from a representative mouse. During approximately half the trials, the active protraction was only a single sweep; in the other traces multiple sweeps were observed. Prolonged periods of active whisking were rare. The periods marked 'I' and 'II' are enlarged in *Figure 1—figure supplement 1A*. (**E**) To indicate the variability in whisker behavior, 100 trials of the same experiment were superimposed. The thick colored line indicates the median. The retraction due to the air-puff is followed by an active protraction. The following supplements are available for *Figure 1*.
DOI: https://doi.org/10.7554/eLife.38852.003

The following source data and figure supplements are available for figure 1:

**Figure supplement 1.** Whisker movements are largely restricted to the period after the air-puff.
DOI: https://doi.org/10.7554/eLife.38852.004

**Figure supplement 2.** Air-puffs induce reflexive whisker movements.
DOI: https://doi.org/10.7554/eLife.38852.005

**Figure supplement 2—source data 1.** Data for *Figure 1—figure supplement 2*.
DOI: https://doi.org/10.7554/eLife.38852.006

---

*2018*; *Bosman et al., 2011*; *Brown and Raman, 2018*; *Kubo et al., 2018*), the dynamics of their responses upon whisker stimulation are heterogeneous (*Brown and Bower, 2001*; *Loewenstein et al., 2005*; *Bosman et al., 2010*; *Chu et al., 2011*). To study the anatomical distribution of these responses within cerebellar lobules crus 1 and crus 2, we mapped the complex spike and simple spike firing of their PCs following ipsilateral whisker pad stimulation with air-puffs in awake mice. Of the 132 single-unit PCs from which we recorded, 118 (89%) showed significant complex spike responses, albeit with large variations in latency and amplitude (*Figure 2A–C*, *Figure 2—*

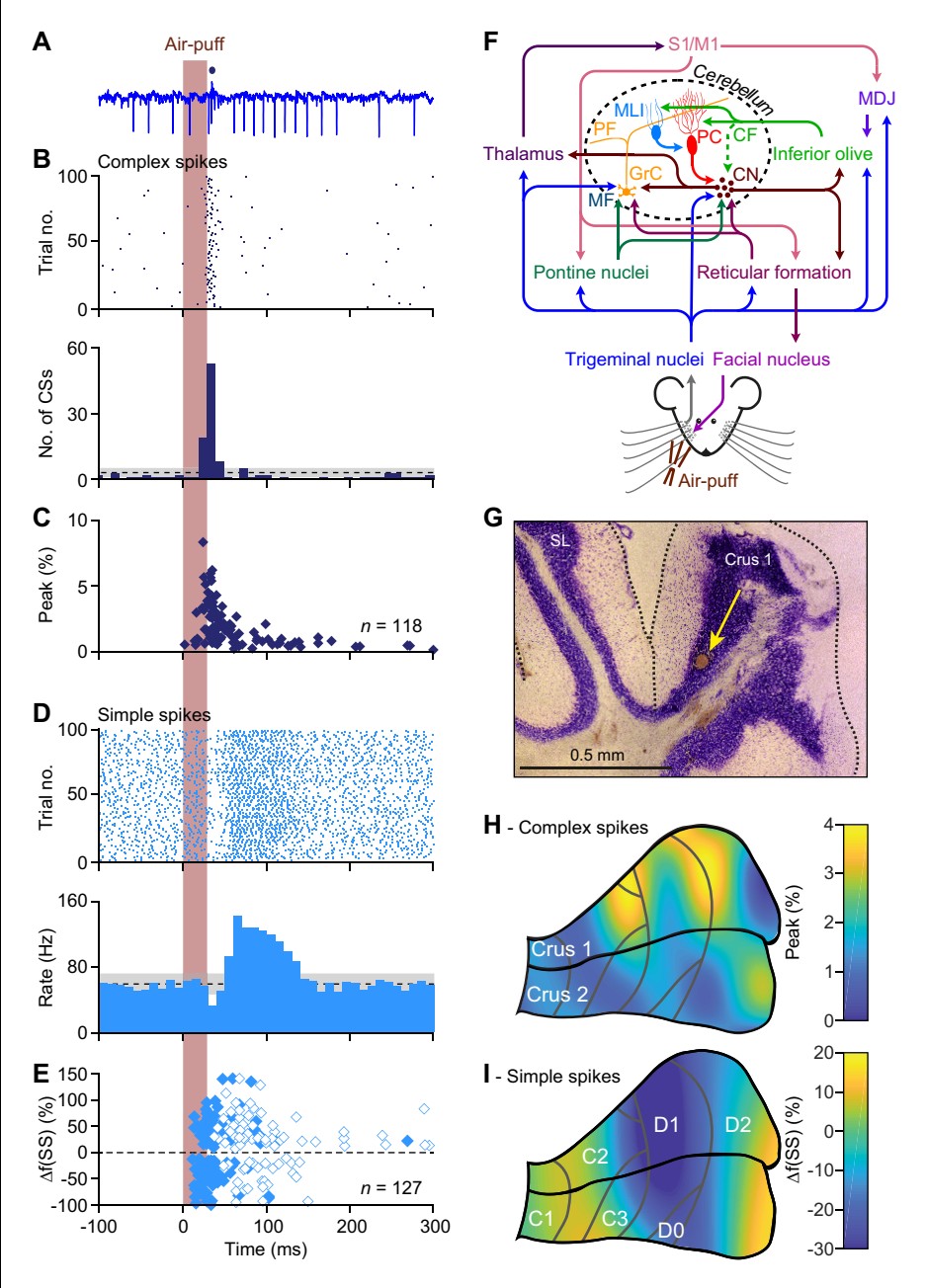

**Figure 2.** Anatomical distribution of Purkinje cell responses to whisker pad stimulation. (**A**) Representative extracellular recording of a cerebellar Purkinje cell (PC) in an awake mouse showing multiple simple spikes (vertical deflections) and a single complex spike that is indicated by a blue dot above the trace. (**B**) Scatter plot and histogram of complex spike firing around the moment of air-puff stimulation of the whisker pad (applied at 0.5 Hz) of the same PC. (**C**) The latencies vs. the peak of the complex spike responses of all 118 PCs with a significant complex spike response. Note that a minority of the PCs showed relatively long latency times. (**D**) Simple spike responses of the same PC showing a bi-phasic response: first inhibition, then facilitation. Note that the simple spike firing frequency of this PC at rest is about 60–70 Hz. (**E**) Peak amplitudes and peak latency times of simple spike responses of all 127 PCs showing a significant simple spike response to whisker pad stimulation. As simple spike responses were often found to be bi-phasic, we represented the first phase with closed and the second phase with open symbols. Non-significant responses are omitted. (**F**) Simplified scheme of the somatosensory pathways from the whisker pad to the PCs and of the motor pathways directing whisker movement. The information flows from the whisker pad via the trigeminal nuclei and the thalamus to the primary somatosensory (**S1**) and motor cortex (**M1**). S1 and M1 project to the inferior olive via the nuclei of the meso-diencephalic junction

*Figure 2 continued on next page*

*Figure 2 continued*

(MDJ) and to the pontine nuclei. Both the inferior olive and the pontine nuclei also receive direct inputs from the trigeminal nuclei. The mossy fibers (MF) from the pontine nuclei converge with direct trigeminal MF and those of the reticular formation on the cerebellar granule cells (GrC) that send parallel fibers (PF) to the PCs. The inferior olive provides climbing fibers (CF) that form extraordinarily strong synaptic connections with the PCs. Both the PFs and the CFs also drive feedforward inhibition to PCs via molecular layer interneurons (MLI). The GABAergic PCs provide the sole output of the cerebellar cortex that is directed to the cerebellar nuclei (CN). The CN sends the cerebellar output both upstream via the thalamus back to the cerebral cortex and downstream to motor areas in the brainstem and spinal cord. The whisker pad muscles are under control of the facial nucleus which is mainly innervated via the reticular formation. Several feedback loops complement these connections. For references, see main text. (G) For most of the PC recordings in this study, the anatomical locations were defined by a combination of surface photographs and electrolytic lesions made after completion of the recordings. An example of such a lesion in crus 1 is shown here in combination with a Nissl staining. SL = simple lobule. (H) Heat map showing the anatomical distribution of the strength of the complex spike responses projected on the surface of crus 1 and crus 2. The locations of all 132 recorded PCs were attributed to a rectangular grid. The average complex spike response strength was calculated per grid position and averaged between each grid position and its neighbor. The grey lines indicate the borders to the cerebellar zones. (I) The same for the variation in the first phase of the simple spike responses. Note that for the simple spikes the blue colors indicate suppression of firing rather than the absence of a response.

DOI: https://doi.org/10.7554/eLife.38852.007

The following figure supplements are available for figure 2:

**Figure supplement 1.** Diversity in Purkinje cell responses.
DOI: https://doi.org/10.7554/eLife.38852.008

**Figure supplement 2.** Anatomy of the whisker region in the cerebellar hemispheres.
DOI: https://doi.org/10.7554/eLife.38852.009

*figure supplement 1A–B*). We considered a response to be significant if it passed the threshold of 3 s.d. above the average of the pre-stimulus interval. Cluster analysis revealed that in terms of complex spike modulation PCs can better be considered as two separate clusters rather than a continuous spectrum (indicated by the lowest absolute BIC value for two components (437, compared to 490 and 442 for one and three components, respectively; *Figure 2—figure supplement 1D*). We refer to the cells of the cluster with the higher complex spike response probability as 'strong' (34%, with a peak response above 1.98%; see Materials and methods) and the other as 'weak' (66%) responders (*Figure 2—figure supplement 1D–F*). Similarly, of the 132 recorded PCs 127 (96%) showed a significant simple spike response. Simple spike responses were often bi-phasic, consisting of a period of inhibition followed by one of excitation, or vice versa (*Figure 2D–E*, *Figure 2—figure supplement 1C*). The trough of the simple spike responses typically correlated in a reciprocal fashion with the peak of the complex spike responses (*Figure 2A–E*; *Figure 2—figure supplement 1A–C*) (*De Zeeuw et al., 2011*; *Zhou et al., 2014*; *Badura et al., 2013*). Only 2 PCs, out of the 132, did not show any significant modulation (i.e. for neither complex spikes nor simple spikes). To chart the spatial distribution of the PCs with different response kinetics upon whisker stimulation we first combined electrolytic lesions (*Figure 2G*) with reconstructions of the electrode entry points, generating a map of the locations of the PCs from which we recorded with the quartz/platinum electrodes ($n$ = 132). Complex spike responses to whisker stimulation were found to be especially strong in parts of crus 1 overlapping with large areas of the C2, D1 and D2 zones (*Figure 2H*), whereas the primary simple spike responses were predominantly facilitating in adjacent areas in the medial and lateral parts of crus 1 and crus 2, as predicted by the overall tendency for reciprocity (*Figure 2H–I*; *Figure 2—figure supplement 1G–H*). This distribution was verified using double-barrel glass pipettes with which we injected the neural tracer, BDA 3000, at the recording spot after recording complex spike responses. Following identification of the source of the climbing fibers in the inferior olive and the projection area in the cerebellar nuclei (*Figure 2—figure supplement 2A–C*), we defined the cerebellar area in which the recorded PC was located (*Apps and Hawkes, 2009*; *Voogd and Glickstein, 1998*). These experiments confirmed that the PCs with strong complex spike responses were situated most prominently in centro-lateral parts of crus 1, whereas the PCs with weak complex spike responses were predominant in adjacent areas in crus 1 and crus 2 (*Figure 2—figure supplement 2D*).

## Large reflexive whisker protractions are preceded by complex spikes

As complex spikes have been reported to be able to encode, at the start of a movement, the destination of arm movements (*Kitazawa et al., 1998*), we wondered whether a similar association could be found for whisker movements. Therefore, we asked whether trials that started with a complex spike involved larger or smaller protractions. To this end, we separated all trials of a session based upon the presence or absence of a complex spike during the first 100 ms after stimulus onset in a single PC. It turned out that during the trials with a complex spike, the protraction was significantly larger (see *Figure 3A* for a single PC; *Figure 3B* for the population of 55 PCs of which we had electrophysiological recordings during whisker tracking and that responded to air-puff stimulation). A direct comparison between the timing of the complex spike response and the difference in whisker position between trials with and without a complex spike revealed that the peak in complex spike activity preceded the moment of maximal difference in position by $63 \pm 4$ ms (mean ±SEM; $n = 55$; *Figure 3C–D*). The maximal difference in protraction in trials with a complex spike equaled 0.80° (median, with IQR of 2.80°; p<0.001), whereas this was only 0.28° (0.92°) for retraction (p=0.002; Wilcoxon matched pairs tests, significant after Bonferroni correction for multiple comparisons: $\alpha = 0.05/3 = 0.017$) (*Figure 3E*). These findings imply that trials that started with a complex spike showed bigger whisker protractions than those without a complex spike.

We next questioned whether there was a correlation between the strength of the complex spike response and the difference in maximal protraction. This did not seem to be the case (R = 0.119; p=0.386; Pearson correlation; *Figure 3—figure supplement 1A*). Thus, in general, the complex spike of any PC showing whisker-related complex spike activity could have a similar predictive power for the amplitude of the subsequent protraction. In line with this, a map showing the distribution of the PCs based upon the correlation of their complex spikes with whisker protraction was fairly homogeneous. Only in an area overlapping with the rostral part of crus 1, a small cluster of PCs was observed whose complex spikes correlated with an unusually large difference in protraction (*Figure 3—figure supplement 1B*). However, since sensory-induced complex spikes were typically more frequent in lateral crus 1, PCs in this area appeared to have overall a stronger correlation with increased touch-induced whisker protraction than the PCs in the surrounding areas (*Figure 3—figure supplement 1C*).

Previous studies showed that motor control can be related to the coherence of complex spike firing of adjacent PCs (*Mukamel et al., 2009*; *Hoogland et al., 2015*). We therefore expected to observe also increased coherence at the trial onsets in our experiments. To test this, we performed two-photon $Ca^{2+}$ imaging to study the behavior of adjacent groups of PCs in crus 1 around the moment of whisker pad air-puff stimulation in awake mice. After injection of the $Ca^{2+}$-sensitive dye Cal-520 we could recognize the dendrites of PCs as parasagittal stripes, each of which showed fluorescent transients at irregular intervals (*Figure 3—figure supplement 2A–B*). Previous studies identified these transients as the result of PC complex spike firing (*Ozden et al., 2008*; *Tsutsumi et al., 2015*; *Schultz et al., 2009*; *De Gruijl et al., 2014*). Occasionally, signals could be found that were shared by many PCs, even in the absence of sensory stimulation (*Figure 3—figure supplement 2B*) in line with earlier reports (*Ozden et al., 2009*; *De Gruijl et al., 2014*; *Mukamel et al., 2009*; *Schultz et al., 2009*). Upon whisker pad stimulation, however, complex spike firing occurred much more often collectively in multiple PCs (*Figure 3—figure supplement 2C*). To quantify this form of coherent firing, we counted the number of complex spikes fired per frame (of 40 ms) and determined the level of coherence using cross-correlation analyses (*Figure 3—figure supplement 2D*) (see also *Ju et al., 2018*). The levels of coherence increased to such strength that they were extremely unlikely to have occurred by the increase in firing frequency alone (compared to a re-distribution of all events based on a Poisson distribution; *Figure 3—figure supplement 2E*). In other words, firing of a single or a few PCs was the dominant mode of activity in the absence of stimulation, and this changed toward the involvement of multiple PCs upon stimulation, firing coherently as can be seen in the change in distribution of coherently active PCs (*Figure 3—figure supplement 2F–G*). We conclude that groups of adjacent PCs respond to whisker pad stimulation by increased complex spike firing with an enhanced level of coherence, which is likely to further facilitate the occurrence of bigger whisker reflexes (see above).

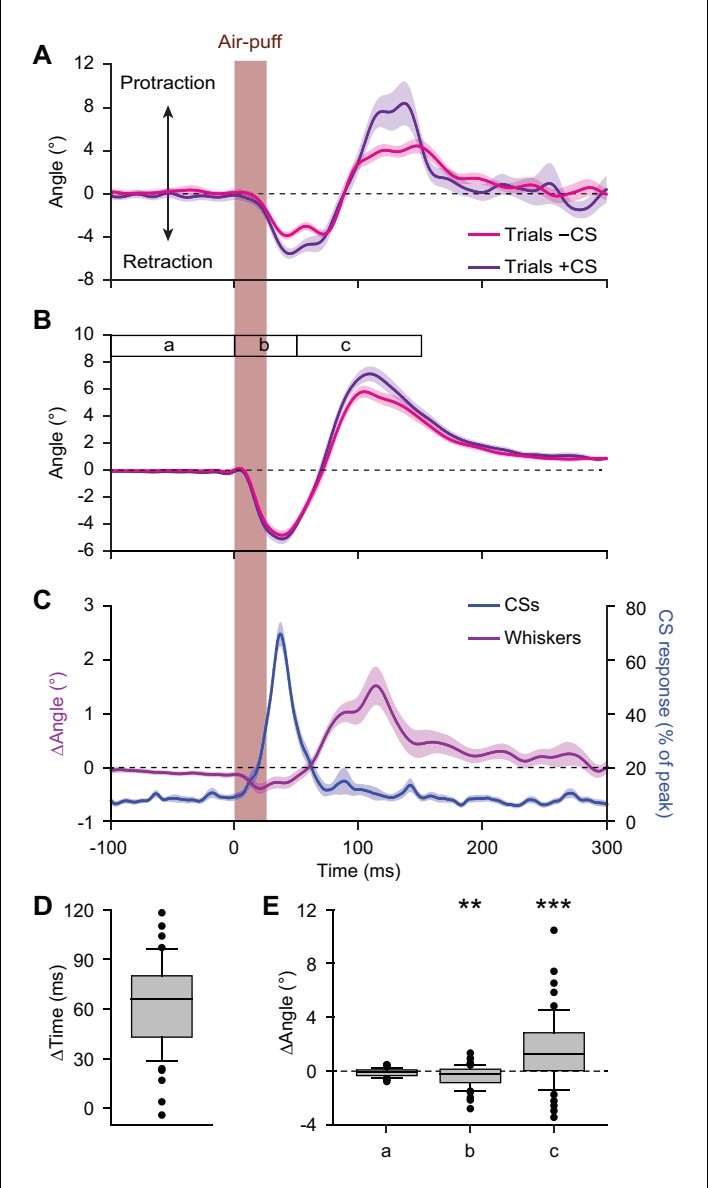

**Figure 3.** Large reflexive whisker protractions are preceded by complex spikes. (**A**) Upon sorting the whisker traces based on the presence (violet) or absence (magenta) of a complex spike (CS) produced by a simultaneously recorded PC in the first 100 ms after stimulus onset, it is apparent that the trials with a complex spike tended to have a stronger protraction. (**B**) This observation was confirmed in the population of PCs with a significant complex spike response to air-puff stimulation ($n$ = 55). (**C**) Averaged convolved peri-stimulus time histograms of complex spikes (*blue*) and the averaged difference in whisker position (*purple*) between trials with and without complex spikes. Complex spikes precede the observed differences in movement. Shaded areas indicate s.d. (**A**) or SEM (**B** and **C**). (**D**) Time intervals between the peak of the complex spike response and the moment of maximal difference in whisker position between trials with and without complex spikes, indicating that the complex spikes lead the whisker movement by approximately 60 ms. (**E**) Changes in average whisker angle before stimulation (period a; see time bar in panel **B**), in maximal retraction (period b) and in maximal protraction (period c) between trials with and without a complex spike in the 100 ms after an air-puff. *$p<0.05$; **$p<0.01$; ***$p<0.001$ (Wilcoxon matched pairs tests (with Bonferroni correction for multiple comparisons in **E**)). See also Source Data file.

DOI: https://doi.org/10.7554/eLife.38852.010

The following source data and figure supplements are available for figure 3:

**Source data 1.** Data for *Figure 3*.
DOI: https://doi.org/10.7554/eLife.38852.013

*Figure 3 continued*

**Figure supplement 1.** Correlation between complex spike firing and whisker protraction is especially strong in the D2 zone.
DOI: https://doi.org/10.7554/eLife.38852.011

**Figure supplement 2.** Coherent complex spike firing is specifically enhanced by whisker pad stimulation.
DOI: https://doi.org/10.7554/eLife.38852.012

## Instantaneous simple spike firing correlates with whisker protraction during reflex

The firing rate of simple spikes has been shown to correlate with whisker position: in the large majority of PCs, simple spike firing is correlated with protraction and in a minority it correlates with retraction (*Brown and Raman, 2018*; *Chen et al., 2016*). This led us to study the correlation in simple spike firing during touch-induced whisker protraction. At first sight, variation in simple spike firing roughly correlated to periods with whisker movement (*Figure 4A–B*). To study this in more detail, we made use of the inter-trial variations in simple spike rate and whisker position, allowing us to make a correlation matrix between these two variables on a trial-by-trial basis (see *ten Brinke et al., 2015*). In a representative example (*Figure 4C*), the whisker protraction and peak in simple spike firing were roughly simultaneous. In the correlation matrix, this is visualized by the yellow color along the 45° line. This turned out to be the general pattern in 25 of the 56 PCs (45%) of which we had electrophysiological recordings during whisker tracking (*Figure 4D*). In all these 25 PCs, there was a positive correlation between instantaneous simple spike firing and whisker protraction that occurred relatively late during the movement, in particular between 80 and 200 ms after the start of the stimulus (*Figure 4C–D*; *Figure 4—figure supplement 1A–C*), thus well after the complex spike responses occurred (*Figure 3C*). In the 31 remaining PCs, that is the ones that did not display a significant correlation when evaluated at the level of individual cells, we still observed a slight, yet significant, correlation at the population level. Remarkably, this correlation was slightly negative, that is possibly reflecting a correlation between simple spike firing and retraction (*Figure 4—figure supplement 1*). We conclude that during the touch-induced whisker reflex simple spikes predominantly correlate with whisker protraction and that this correlation is maximal without a clear time lead or lag, unlike the complex spikes, the occurrence of which tended to precede the reflexive protraction.

## 4 Hz air-puff stimulation leads to acceleration of simple spike response and to stronger protraction of whiskers

Next, we investigated whether sensory experience could modulate the touch-induced whisker protraction, the frequency of simple spike firing and the relation between them. We hypothesized that whisker movements might be enhanced following air-puff stimulation at 4 Hz, as this frequency has been shown to be particularly effective in inducing potentiation at the parallel fiber-to-PC synapse (*Coesmans et al., 2004*; *D'Angelo et al., 2001*; *Lev-Ram et al., 2002*; *Ramakrishnan et al., 2016*). Indeed, application of this 4 Hz air-puff stimulation to the whisker pad for only 20 s was sufficient to induce an increase in the maximal protraction (average increase $17.9 \pm 3.9\%$; mean $\pm$SEM; p<0.001; Wilcoxon-matched pairs test; $n = 16$ mice) (*Figure 5A–B*; *Supplementary file 1B*).

This change in the amplitude of the touch-induced whisker protraction was not accompanied by any substantial change in the complex spike response to whisker pad stimulation (p=0.163; Wilcoxon matched pairs test; $n = 55$ PCs) (*Figure 5C*; *Supplementary file 1B*). However, the rate of simple spike firing upon air-puff stimulation was markedly increased after 20 s of 4 Hz air-puff stimulation. This was especially clear during the first 60 ms after the air-puff (p=0.003; Wilcoxon matched pairs test; $n = 55$ PCs) (*Figure 5D*; *Supplementary file 1B*). Overlaying the averaged whisker traces and PC activity profiles highlighted the earlier occurrence of facilitation in simple spike firing after the 4 Hz air-puff stimulation protocol (*Figure 5E*). To study this timing effect in more detail, we repeated the trial-based correlation analysis (cf. *Figure 4C–D*). The short period of 4 Hz air-puff stimulation caused an anticipation of the moment of maximal correlation between simple spike firing and whisker position. Along the 45° line – thus regarding only the zero-lag correlation between simple spike firing and whisker position – this changed from $152.1 \pm 18.1$ ms to $90.7 \pm 9.4$ ms (means $\pm$ SEM); p=0.020; $t = 2.664$; df = 13; paired $t$ test; $n = 14$ PCs) (*Figure 5—figure supplement 1A–B*). The

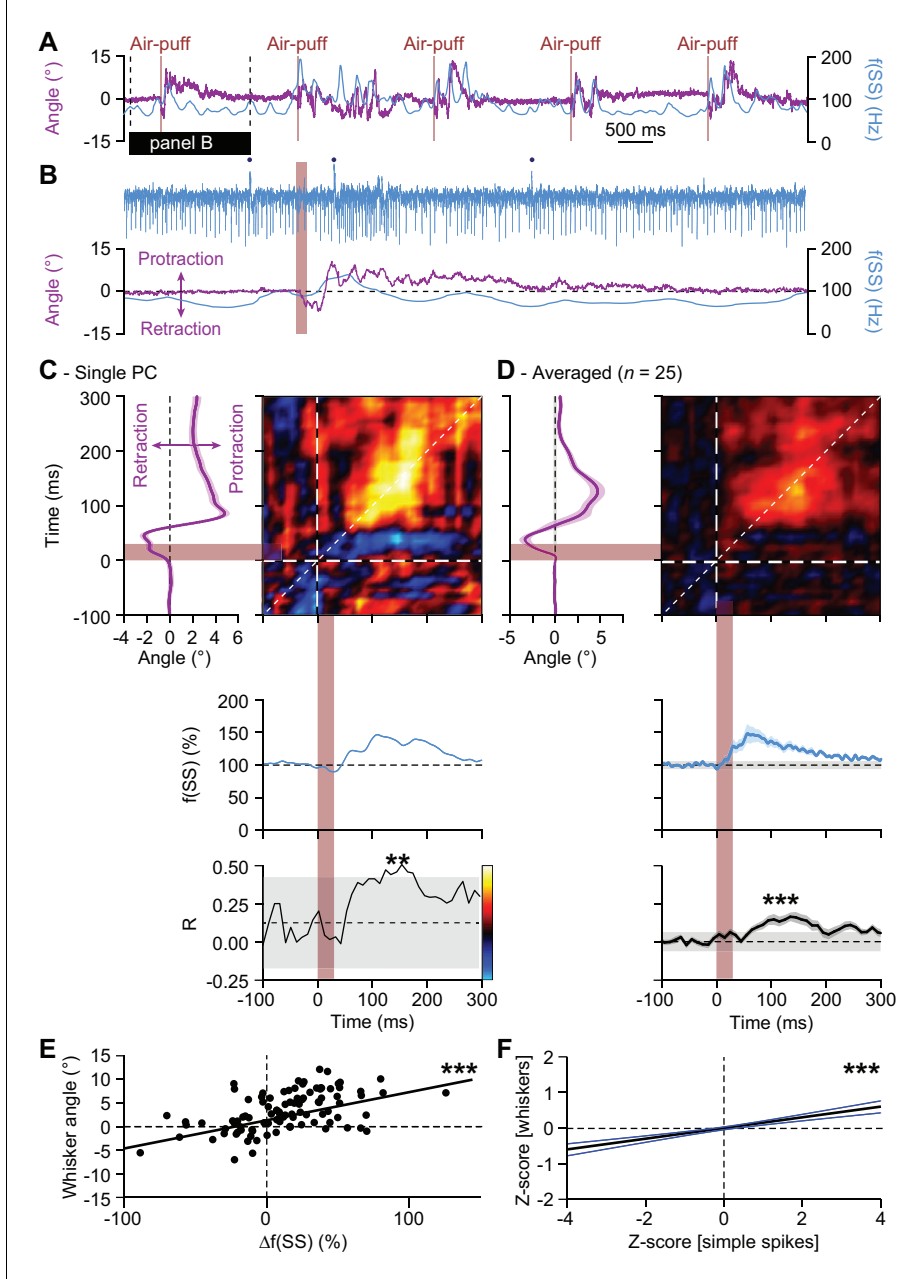

**Figure 4.** Instantaneous simple spike firing correlates with whisker protraction during reflex. (**A**) Changes in the instantaneous simple spike (SS) firing rate (convolved with a 6 ms Gaussian kernel; *blue*) correlate roughly with whisker movement (*purple*). This is illustrated with a representative recording of a PC. Vertical brown lines indicate the moments of air-puff stimulation to the (ipsilateral) whisker pad. The horizontal black line designates the interval expanded in (**B**). Blue dots mark complex spikes. (**C**) Correlation matrix showing a clear positive correlation of simple spike firing (blue trace at the bottom shows convolved peri-stimulus time histogram triggered on air-puff stimulation) and whisker protraction (red trace at the left; indicated is the mean ±SEM of the whisker position) based on a trial-by-trial analysis. The correlation coefficient (**R**) over the dashed 45° line is shown at the bottom, together with the 99% confidence interval (*grey area*). These data correspond to the example PC shown in (**A–B**). Averaged data from all 25 PCs that displayed a significant correlation between simple spike rate and whisker position are shown in (**D-E**). Scatter plots with linear regression lines show a positive correlation between whisker protraction and instantaneous simple spike firing as illustrated here for the PC represented in (**C**) (R = 0.517; p<0.001; Pearson correlation). Data are taken from the moment with the strongest correlation (150–160 ms after the onset of the air-puff for both parameters). (**F**) For all PCs with a significant correlation between whisker angle and simple spike rate, this correlation turned out to be positive when evaluating 100 trials for each of the 25

*Figure 4 continued on next page*

*Figure 4 continued*

Purkinje cells (R = 0.199; p<0.001; Pearson correlation). Shown a the linear regression line (*black*) and the 95% confidence intervals (*blue*). The experiments are normalized based upon their Z-score. Data are taken from the moment with the strongest correlation (120–130 ms (whiskers) vs. 140–150 ms (simple spikes)). Thus, increased simple spike firing correlates with whisker protraction. **p<0.01; ***p<0.001.

DOI: https://doi.org/10.7554/eLife.38852.014

The following figure supplement is available for figure 4:

**Figure supplement 1.** Simple spike firing is predominantly associated with protraction.

DOI: https://doi.org/10.7554/eLife.38852.015

slope of the correlation between the instantaneous simple spike frequency and the whisker position remained unaltered (p=0.197, *t* = 1.360, df = 13, *n* = 14, paired *t* test) (*Figure 5—figure supplement 1C–D*). However, the point of maximal correlation was no longer with a zero-lag, but after induction the simple spikes led the whisker position (pre-induction: Δtime = 0 ± 10 ms; post-induction: Δtime = 20 ± 30 ms; medians ± IQR; *n* = 14; p=0.001; Wilcoxon matched-pairs test) (*Figure 5F*). Thus, not only the simple spike rate increased, but also its relative timing to the touch-induced whisker protraction changed, now preceding the likewise increased touch-induced whisker protraction.

During the entrainment itself (i.e. during the 20 s period with 4 Hz air-puff stimulation), the whisker responses as well as the complex spike and the simple spike responses to each air-puff were weakened compared to the pre-induction period during which we used 0.5 Hz stimulation. More specifically, the touch-induced whisker protraction decreased by 62.2% (median; IQR = 37.5%). The maximum response of the complex spikes significantly decreased from a median of 1.27% (with an IQR of 1.89%) during pre-induction to 0.52% (with an IQR of 0.43%) during induction (p<0.001; Wilcoxon matched pairs tests, *n* = 55 PCs), and the average modulation of the simple spikes in the first 200 ms after the puff decreased from a median of 5.9% (with an IQR of 18.2%) during pre-induction to -0.3% (IQR = 3.13) during induction (p=0.039, Wilcoxon matched-pairs test) (*Figure 5—figure supplement 2*). Thus, during the 4 Hz training stage, all responses – both at the behavioral and neuronal level – diminished compared to the preceding 0.5 Hz stimulation stage.

Given the correlation between instantaneous simple spike rate and whisker position described above, one would expect that contralateral air-puff stimulation – which triggers a stronger protraction (*Figure 1—figure supplement 2*) – also triggers a stronger simple spike response. To test this hypothesis, we recorded PC activity while stimulating the ipsi- and contralateral whiskers in a random sequence (*Figure 5—figure supplement 3A–B*). The change in maximal protraction was considerable (difference in maximal protraction: 7.30 ± 1.24° (mean ±SEM); *n* = 9 mice) (*Figure 5—figure supplement 3C*; cf. *Figure 1—figure supplement 2E*). Possibly, the absence of the direct mechanical retraction on the ipsilateral side during contralateral air-puff stimulation can explain part of this difference, which is also in line with the earlier onset of the protraction during contralateral stimulation (*Figure 5—figure supplement 3C*). However, in addition a change in simple spikes may contribute to this difference as well, as the simple spikes increased significantly more during contralateral stimulation (increase during first 60 ms after air-puff onset for contra- vs. ipsilateral stimulation: 13.7 ± 5.3%; mean ±SEM; p=0.023; *t* = 2.413; df = 26; paired *t* test; *n* = 27 PCs) (*Figure 5—figure supplement 3E*). Such a contribution is compatible with the fact that most mossy fiber pathways related to whisker movement are bilateral with a contralateral preponderance (*Bosman et al., 2011*). Instead, the complex spikes were less activated during contralateral stimulation (complex spike peak response: ipsilateral: 1.40% (1.25%); contralateral: 0.71% (0.81%); medians (IQR); p<0.001; Wilcoxon matched-pairs test; *n* = 27 PCs) (*Figure 5—figure supplement 3D*). This response is in line with a bilateral component of the projection from the trigeminal nucleus to the olive (*De Zeeuw et al., 1996*).

To establish a causal link between increases in simple spike firing and whisker protraction, artificial PC stimulation would also have to affect whisker movement. Previously, it has been shown that simple spikes modulate ongoing whisker movements rather than initiate them (*Brown and Raman, 2018*; *Chen et al., 2016*; *Proville et al., 2014*). To find out whether simple spike firing could modulate touch-induced whisker protraction under our recording conditions, we investigated the impact of activation of PCs by optogenetic stimulation. To this end we used *Pcp2-Ai27* mice, which express

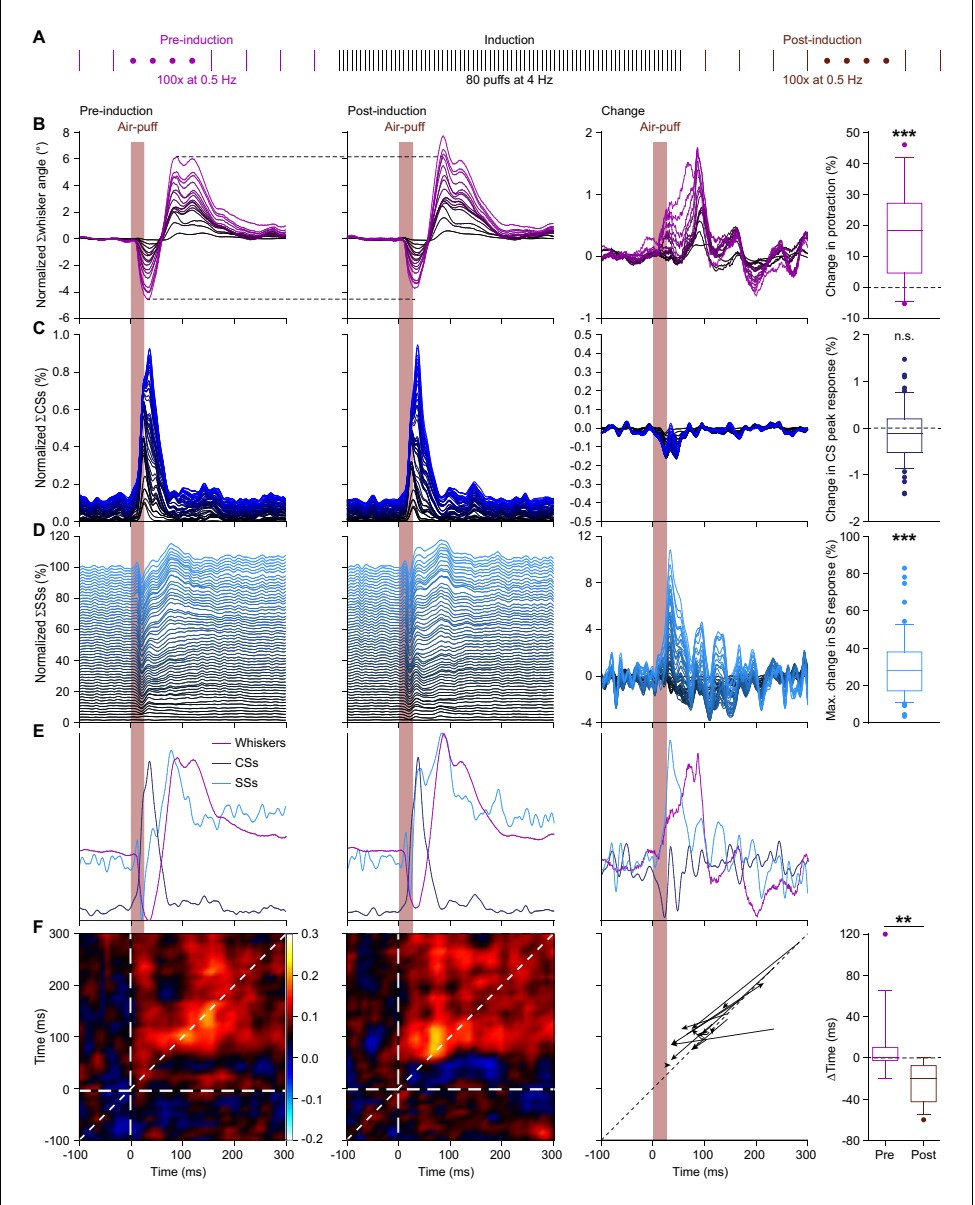

**Figure 5.** 4 Hz air-puff stimulation leads to acceleration of the simple spike response and to stronger protraction of the whiskers. (A) Induction protocol: air-puff stimulation at 0.5 Hz is used to characterize the impact of a brief period (20 s) of 4 Hz air-puff stimulation. (B) Stacked line plots showing the averaged whisker responses before (1st column) and after (2nd column) 4 Hz air-puff stimulation. The plots are sorted by the increase of the touch-induced whisker protraction (3rd column). Each color depicts one mouse. Plots are normalized so that the most intense color represents the average of 16 mice. 4 Hz air-puff stimulation leads to a stronger touch-induced whisker protraction (4th column). Similar plots for complex spikes (C), showing little change, and simple spikes (D), showing a clear increase in firing, especially during the early phase of the response. For comparison, the averages are superimposed in (E) (for y-scaling and variations refer to B–D). Trial-by-trial analysis of 14 Purkinje cells before and after 4 Hz air-puff stimulation (cf. *Figure 4C–D*) highlighting the anticipation of simple spike firing (F). The x-axis is based upon the instantaneous simple spike firing frequency and the y-axis upon the whisker angle. The moment of maximal correlation between simple spike firing and whisker movement anticipated after induction, as can be seen by the change in position of the yellow spot between the correlation plots in the 1st and 2nd column (see also *Figure 5—figure supplement 1A–B*). After induction, the maximal correlation implied a lead of the simple spikes, as illustrated for each PC in the graph of the 3rd column. Every arrow indicates the shift of the position of the maximal correlation between before and after induction. Overall, the difference in timing between the maximal

*Figure 5 continued on next page*

*Figure 5 continued*

correlation changed from around 0 ms pre-induction to an approximate lead of 20 ms of the simple spikes after induction (4[th] column). **p<0.01; ***p<0.001. See also *Supplementary file 1B* and Source Data File.

DOI: https://doi.org/10.7554/eLife.38852.016

The following source data and figure supplements are available for figure 5:

**Figure supplement 1—Source data 1.** Data for *Figure 5—figure supplement 1*.
DOI: https://doi.org/10.7554/eLife.38852.018
**Source data 1.** Data for *Figure 5*.
DOI: https://doi.org/10.7554/eLife.38852.025
**Figure supplement 1.** Simple spike response anticipates after 4 Hz air-puff stimulation.
DOI: https://doi.org/10.7554/eLife.38852.017
**Figure supplement 2.** Purkinje cell responses during 4 Hz air-puff stimulation.
DOI: https://doi.org/10.7554/eLife.38852.019
**Figure supplement 2—source data 1.** Data for *Figure 5—figure supplement 2*.
DOI: https://doi.org/10.7554/eLife.38852.020
**Figure supplement 3.** Contralateral whisker pad stimulation induces stronger whisker protraction and stronger simple spike responses.
DOI: https://doi.org/10.7554/eLife.38852.021
**Figure supplement 3—source data 1.** Data for *Figure 5—figure supplement 3*.
DOI: https://doi.org/10.7554/eLife.38852.022
**Figure supplement 4.** Optogenetic stimulation of Purkinje cells increases whisker protraction following air-puff stimulation.
DOI: https://doi.org/10.7554/eLife.38852.023
**Figure supplement 4—source data 1.** Data for *Figure 5—figure supplement 4*.
DOI: https://doi.org/10.7554/eLife.38852.024

channelrhodopsin-2 exclusively in their PCs and which respond with a strong increase in their simple spike firing upon stimulation with blue light (*Witter et al., 2013*). We placed an optic fiber with a diameter of 400 µm over the border between crus 1 and crus 2 and compared air-puff induced whisker movements among randomly intermingled trials with and without optogenetic PC stimulation. The period of optogenetic stimulation (i.e. 100 ms) was chosen to mimic preparatory activity of PCs and thus corresponded well to the period during which we observed increased simple spike firing after 4 Hz air-puff stimulation (*Figure 5D*). As expected, the whisker protraction was substantially bigger during the period of optogenetic stimulation (p<0.001; $t = 4.411$; df = 12; paired $t$ test; $n = 13$ mice; *Figure 5—figure supplement 4*). Thus, even though optogenetic stimulation of PCs can also trigger secondary feedback mechanisms that may influence the outcome (*Witter et al., 2013*; *Chaumont et al., 2013*), we conclude that increases in simple spike firing can cause stronger whisker protraction.

## Complex spikes inhibit increased simple spike firing

As cerebellar plasticity is bi-directional and under control of climbing fiber activity (*Ohtsuki et al., 2009*; *Lev-Ram et al., 2003*; *Coesmans et al., 2004*), we wanted to find out to what extent plastic changes in simple spike activity can be related to the strength of the complex spike response generated by climbing fibers. To this end we compared for each PC the strengths of the complex spike and simple spike responses before, during and after the 4 Hz air-puff stimulation. As expected, we found a significant negative correlation between the strength of the complex spike response, as measured by the peak of the PSTH before the 4 Hz air-puff stimulation, and the change in simple spike response following this 4 Hz stimulation (R = 0.311; p=0.021; Pearson correlation; $n = 55$ PCs) (*Figure 6A*). We further substantiated these findings by looking separately at the average complex spike firing frequency of the strong and weak responders (cf. *Figure 2—figure supplement 1E*). The correlation found between the frequency of complex spike firing and the change in simple spike activity after 4 Hz air-puff stimulation proved to be present only in the weak responders, taking the firing rate during the pre-induction and induction period into account (*Figure 6—figure supplement 1*). This is again in line with the notion that parallel fiber activity in the absence of climbing fiber activity promotes parallel fiber to PC LTP (*Coesmans et al., 2004*; *Lev-Ram et al., 2002*;

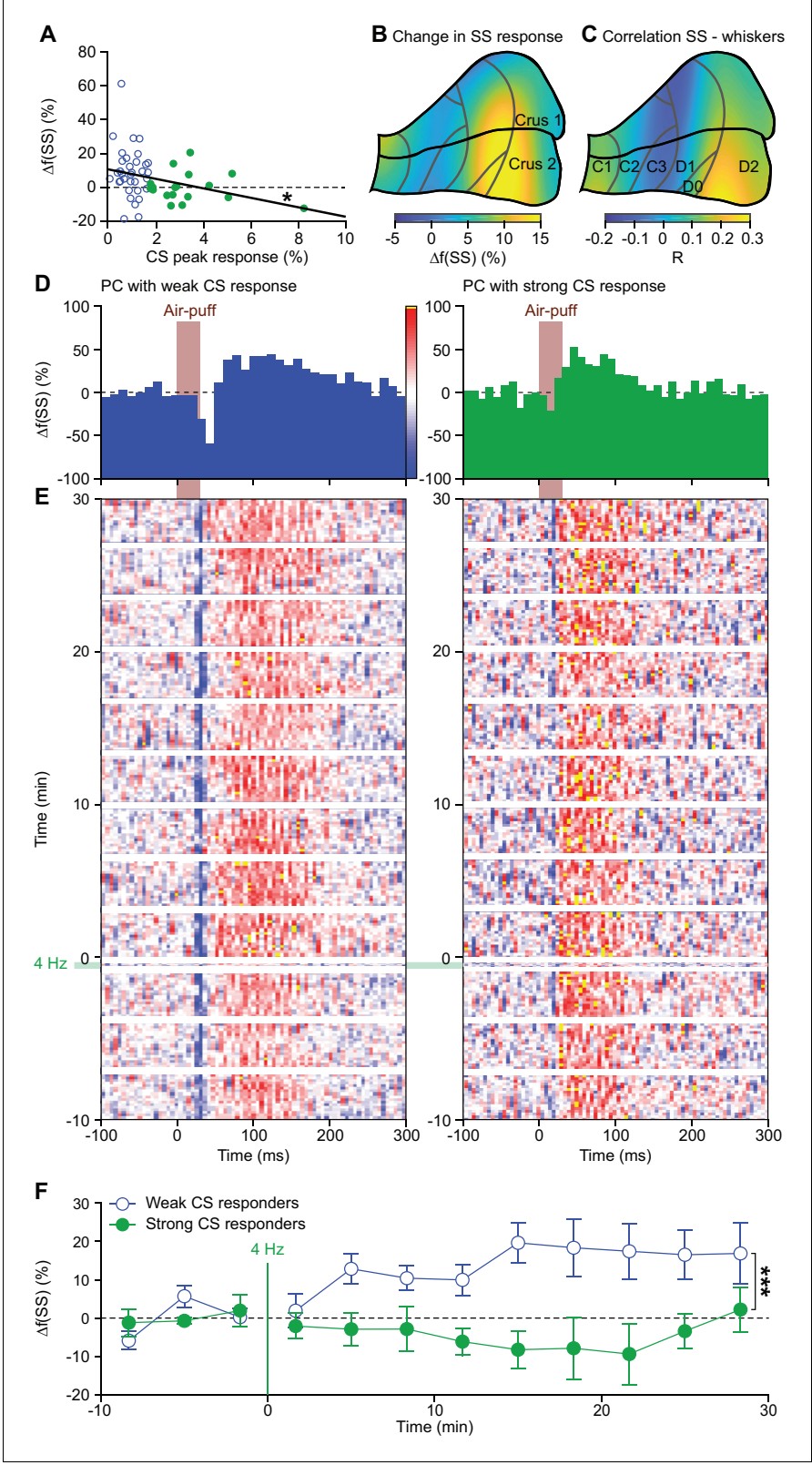

**Figure 6.** Complex spikes inhibit increased simple spike firing. (**A**) Repeated sensory stimulation induced an increase in simple spike (SS) response to whisker pad stimulation (see *Figure 5*). This increase in simple spike responses was, however, not observed in all PCs: there was a clear negative correlation between the strength of the complex spike (CS) response and the potentiation of the simple spike response. Overall, the simple spike

*Figure 6 continued on next page*

*Figure 6 continued*

potentiation was larger in the PCs with a weak complex spike response than in those with a strong complex spike response (cf. *Figure 2—figure supplement 1*). (B) Heat map showing the anatomical distribution of the strength of the simple spike increase projected on the surface of crus 1 and crus 2. The 55 PCs were attributed to a rectangular grid. The average simple spike response strength was calculated per grid position and averaged between each grid position and its neighbor. The grey lines indicate the borders to the cerebellar zones (see *Figure 2—figure supplement 2D*). (C) Heat map of the distribution of PCs cells based upon the correlation of their simple spike rate and whisker position (cf. *Figure 4D*). Note that the strongest increase of simple spike responses after 4 Hz air-puff stimulation occurred in the region that also displayed the strongest correlation between instantaneous simple spike rate and whisker position. (D) Example PSTHs of the simple spike response to whisker pad air-puff stimulation of representative PCs and how they changed over time, depicted as heat maps of the instantaneous simple spike frequency (E); see scale bar in (D). The left column displays the data from a PC with a weak complex spike response, the right column of one with a strong complex spike response. The induction period is indicated with '4 Hz'. (F) The number of simple spikes following an air-puff stimulation increased in weakly responding Purkinje cells and this increase remained elevated until the end of the recording (at least 30 min). In contrast, this increase was not found in Purkinje cells with strong complex spike responses. *p<0.05; **p<0.01; ***p<0.001.

DOI: https://doi.org/10.7554/eLife.38852.026
The following figure supplements are available for figure 6:

**Figure supplement 1.** Complex spike rates are negatively correlated with sensory-induced potentiation.
DOI: https://doi.org/10.7554/eLife.38852.027
**Figure supplement 2.** 4 Hz air-puff stimulation enhances reflexive whisker protraction for at least 30 min.
DOI: https://doi.org/10.7554/eLife.38852.028

*Ramakrishnan et al., 2016*). The PCs with the strongest effect of 4 Hz air-puff stimulation on simple spike firing were mainly located in the lateral part of crus 2 (*Figure 6B*), posterior to the crus 1 area with the strongest complex spike responses (*Figure 2H*). We compared the location of this lateral crus 2 area to that of the PCs with the strongest correlations between simple spike firing and whisker protraction and we found these two crus 2 locations to match well (*Figure 6B–C*).

The impact of 4 Hz air-puff stimulation on the simple spike activity of PCs with a weak complex spike response lasted as long as our recordings lasted (i.e. at least 30 min), whereas that on PCs with a strong complex spike response was not detectable during this period (*Figure 6D–F*). Indeed, the weak responders differed significantly from the strong responders in this respect (weak vs. strong responders: p=0.005; $F = 3.961$; df = 4.424; two-way repeated measures ANOVA with Greenhouse-Geyser correction; $n = 8$ weak and $n = 6$ strong responders; *Figure 6F*). Likewise, the impact of the 4 Hz air-puff stimulation on touch-induced whisker protraction also lasted throughout the recording in that the protraction sustained (*Figure 6—figure supplement 2*). Thus, both simple spikes and whisker muscles remained affected by the 4 Hz air-puff stimulation for as long as our recordings lasted.

## Expression of PP2B in Purkinje cells is required for increased protraction and simple spike firing following 4 Hz air-puff stimulation

In reduced preparations, 4 Hz stimulation of the parallel fiber inputs leads to LTP of parallel fiber to PC synapses (*Coesmans et al., 2004*; *Lev-Ram et al., 2002*; *Ramakrishnan et al., 2016*). At the same time, parallel fiber LTP is inhibited by climbing fiber activity (*Coesmans et al., 2004*; *Lev-Ram et al., 2003*; *Ohtsuki et al., 2009*). Hence, our data appear in line with a role for parallel fiber LTP as a potential mechanism underlying the observed increase in simple spike firing upon a brief period of 4 Hz stimulation. To further test a potential role for LTP, we repeated our 4 Hz air-puff stimulation experiments in *Pcp2-Ppp3r1* mice, which lack the PP2B protein specifically in their PCs, rendering them deficient of parallel fiber-to-PC LTP (*Schonewille et al., 2010*) (*Figure 7A*). The impact of 4 Hz air-puff stimulation on the maximal protraction was significantly less in the *Pcp2-Ppp3r1* mutant mice compared to wild types (p=0.044, $t = 2.162$, df = 19, $t$ test; *Figure 7B–D*). Accordingly, in contrast to those in their wild-type (WT) littermates (p<0.001, $t = 4.122$, df = 15, $t$ test), the maximal touch-induced whisker protraction before and after induction was not significantly different in *Pcp2-Ppp3r1* mice (p=0.647, $t = 0.470$, df = 12, $t$ test; *Figure 7E*). Thus, *Pcp2-Ppp3r1* mice do not show increased touch-induced whisker protraction after 4 Hz air-puff stimulation.

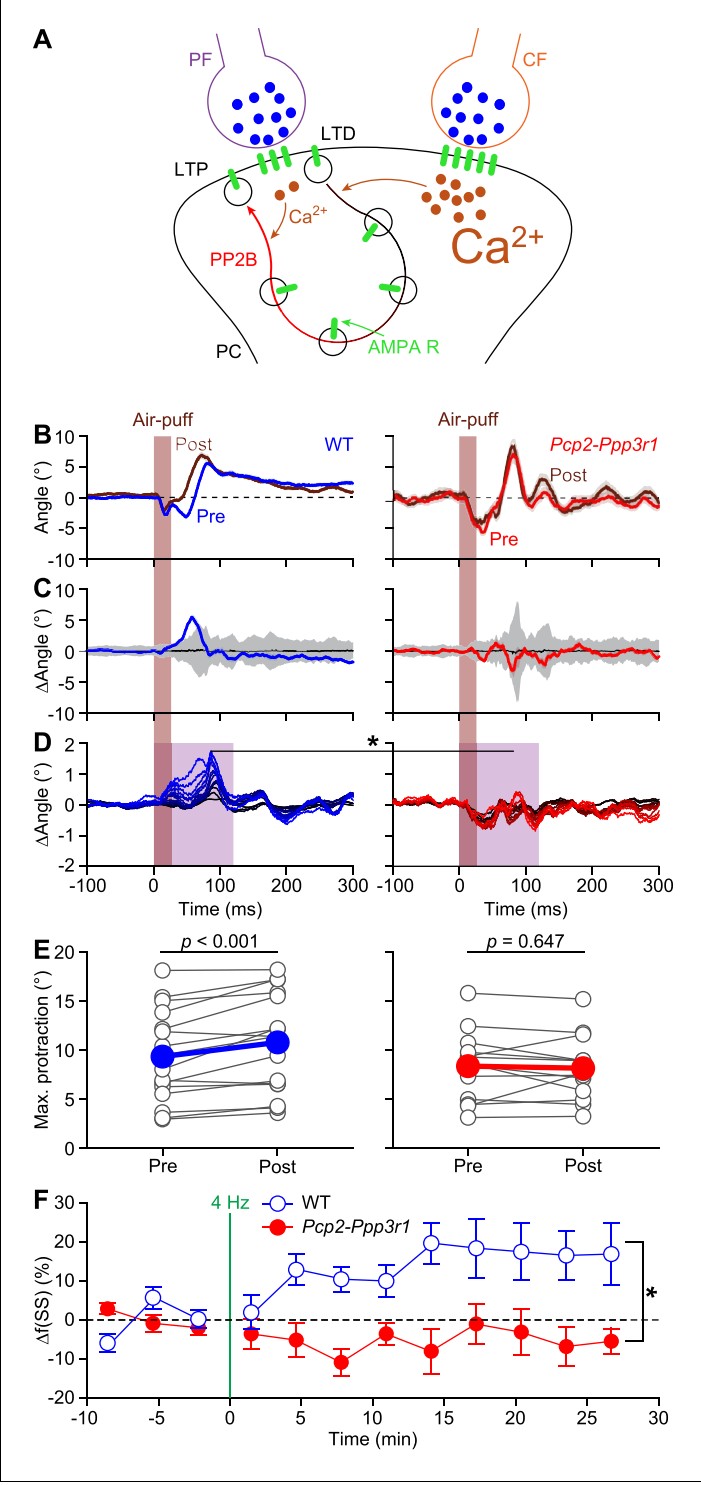

**Figure 7.** Expression of PP2B in Purkinje cells is required for increased protraction and simple spike firing following 4 Hz air-puff stimulation. (**A**) Schematic representation of the principal pathways regulating bidirectional plasticity at the parallel fiber (PF) to Purkinje cell (PC) synapses. The direction of PF-PC plasticity depends on the intracellular $Ca^{2+}$ concentration ($[Ca^{2+}]_i$) that is largely determined by climbing fiber (CF) activity. Following CF activity, $[Ca^{2+}]_i$ raises rapidly and activates a phosphorylation cascade involving α-$Ca^{2+}$/calmodulin-dependent protein kinase II (CaMKIIA) and several other proteins eventually leading to internalization of AMPA receptors and consequently to long-term depression (LTD). PF volleys in the absence of CF activity, on the other hand, result in a moderate increase in $[Ca^{2+}]_i$, activating a protein cascade involving protein phosphatase 2B (PP2B, encoded by

*Figure 7 continued on next page*

*Figure 7 continued*

*Ppp3r1*) that promotes the insertion of new AMPA receptors into the postsynaptic density, thereby leading to long-term potentiation (LTP) of the PF-PC synapse. GluA3 subunits are part of the postsynaptic AMPA receptors. (B) Example of a representative mouse with the averaged whisker movements before and after 4 Hz air-puff stimulation, showing a stronger protraction afterwards, as evidenced by the differences between post- and pre-induction compared to a bootstrap analysis on the normal variation in whisker movements (C); shade: 99% confidence interval). Variations in whisker protraction in *Pcp2-Ppp3r1* mutants did generally not exceed the expected variability (right). (D) Stacked line plots of whisker movement differences between post- and pre-induction for all mice highlighting the absence of increased touch-induced whisker protraction in *Pcp2-Ppp3r1* mutant mice. The plots are normalized so that the brightest line indicates the average per genotype (*n* = 16 WT and *n* = 13 *Pcp2-Ppp3r1* KO mice). (E) The average maximal protraction before and after induction for each mouse confirms the increase in WT, but not in *Pcp2-Ppp3r1* mutant mice. The colored symbols indicate the average per genotype. (F) In contrast to simple spike responses in WT mice, those in *Pcp2-Ppp3r1* KO mice could not be potentiated by our 4 Hz air-puff stimulation protocol. This effect was stable, also during longer recordings. For this analysis, we selected those with weak complex spike responses, as the PCs with a strong complex spike response did not show increased simple spike firing after four air-puff stimulation (see ***Figure 6A***). *p<0.05; **p<0.01; ***p<0.001.

DOI: https://doi.org/10.7554/eLife.38852.029

The following figure supplement is available for figure 7:

**Figure supplement 1.** Also in *Pcp2-Ppp3r1* KO mice, complex spike rates are negatively correlated with sensory-induced potentiation.

DOI: https://doi.org/10.7554/eLife.38852.030

In line with the absence of increased touch-induced whisker protraction, also the increase in simple spike firing observed in wild type mice was absent in *Pcp2-Ppp3r1* mice. As the strong complex spike responders in WTs did not show changes in simple spike activity (cf. ***Figure 6***), we compared weak complex spike responders of both genotypes. Simple spike responses were stably increased in WT PCs with a weak complex spike response following 4 Hz air-puff stimulation (as shown in ***Figure 6F***), but not in those of *Pcp2-Ppp3r1* mice (effect of genotype: p=0.003, *F* = 4.361, df = 4.137, two-way repeated measures ANOVA with Greenhouse-Geyser correction; *n* = 8 WT and *n* = 9 *Pcp2-Ppp3r1* PCs) (***Figure 7F***). Despite the lack of potentiation, we found that the *Pcp2-Ppp3r1* mice still had a significant correlation between the complex spike frequency during the induction block and changes in simple spike activity (R = 0.489, p=0.013, Pearson correlation; ***Figure 7—figure supplement 1A***); this correlation may result from other forms of plasticity that are still intact in *Pcp2-Ppp3r1* mice (***Schonewille et al., 2010***). Yet, in line with the absence of increased simple spike responsiveness, the correlation between changes in simple spike firing during the induction block and the impact of 4 Hz air-puff stimulation, as present in the WT PCs, was absent in the *Pcp2-Ppp3r1* mice (***Figure 7—figure supplement 1B***). Thus, in the absence of the PP2B protein in PCs, the impact of 4 Hz air-puff stimulation on touch-induced whisker protraction as well as on the simple spike responsiveness was not detectable. These correlations between complex spike and simple spike firing on the one hand and modification of the simple spike response to whisker pad stimulation on the other hand further strengthen our hypothesis that parallel fiber to PC LTP is one of the main mechanisms that underlies the long-term changes that can be observed at both the level of simple spike activity and whisker protraction after 4 Hz air-puff stimulation.

## Expression of AMPA receptor GluA3 subunits in Purkinje cells is required for increased protraction and simple spike firing following 4 Hz air-puff stimulation

To control for compensatory mechanisms specific for *Pcp2-Ppp3r1* mice we used a second, independent, yet also PC-specific, mutant mouse line deficient in parallel fiber LTP. In these mice (*Pcp2-Gria3*), PCs lack the AMPA receptor GluA3 subunit (***Gutierrez-Castellanos et al., 2017***). As in the *Pcp2-Ppp3r1* mice, we did not find evidence for increased whisker protraction after 4 Hz air-puff stimulation (e.g., change in whisker angle during the first 120 ms after air-puff onset: WT vs. *Pcp2-Gria3* mice: p=0.007, Tukey's post-hoc test after ANOVA (p=0.001, *F* = 9.111, df = 2), *n* = 16 WT and *n* = 6 *Gria3* deficient mice) (***Figure 8A–C***). Moreover, as in the *Pcp2-Ppp3r1* mice, also the increase in simple spike responsiveness after 4 Hz stimulation was absent in *Pcp2-Gria3* mice

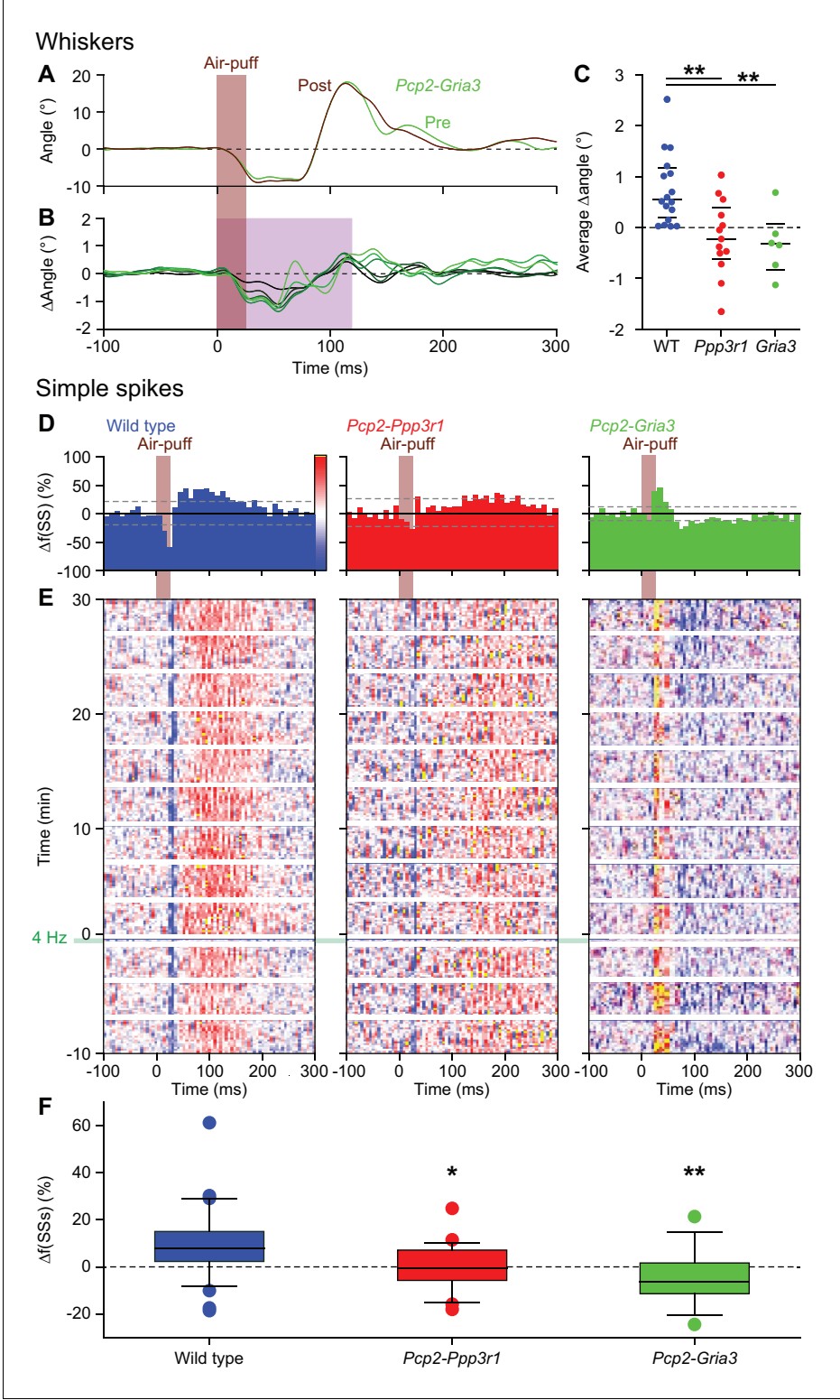

**Figure 8.** Expression of AMPA receptor GluA3 subunits in Purkinje cells is required for increased protraction and simple spike firing following 4 Hz air-puff stimulation. (**A**) Example of a representative *Pcp2-Gria3* mutant mouse that is deficient for the GluA3 subunit with the averaged whisker movements before and after 4 Hz air-puff stimulation, showing similar degrees of protraction. (**B**) Overall, 4 Hz air-puff stimulation did not result in stronger whisker protraction in *Pcp2-Gria3* mutant mice as observed in WT mice (see *Figure 7*). This is illustrated with a stacked line plot. (**C**) Comparison of the average change in whisker angle over the 120 ms following the onset of

*Figure 8 continued on next page*

*Figure 8 continued*

the air-puff shows enhanced protraction in WT (*n* = 16), but not in LTP-deficient mice - neither in *Pcp2-Ppp3r1* (*n* = 13) nor in *Pcp2-Gria3* (*n* = 6) mutants, pointing towards a central role for parallel fiber-to-Purkinje cell LTP for the enhanced protraction in WT mice following a brief period of 4 Hz air-puff stimulation. The horizontal lines indicate the medians and the 1st and 3rd quartiles. The lack of change in whisker protraction following 4 Hz air-puff stimulation was reflected in the lack of change in simple spike responses as illustrated in three representative PCs (cf. *Figure 6D–E*). On top are the peri-stimulus time histograms (D) followed by heat maps illustrating the instantaneous firing rate over time (E). The induction period is indicated with '4 Hz'. (F) Overall, WT PCs (*n* = 35) showed increased simple spike firing after 4 Hz stimulation, while those in *Pcp2-Ppp3r1* (*n* = 21) or *Pcp2-Gria3* (*n* = 13) mutant mice did not. For this analysis, we restricted ourselves to the PCs with weak complex spike responses as the PCs with strong complex spike responses did not show potentiation in the WT mice (see *Figure 6A*) and to the first 100 trials after induction. *p<0.05; **p<0.01. See also Source Data File.
DOI: https://doi.org/10.7554/eLife.38852.031

The following source data and figure supplements are available for figure 8:

**Figure supplement 1—Source data 1.** Data for *Figure 8—figure supplement 1*.
DOI: https://doi.org/10.7554/eLife.38852.033

**Figure supplement 2—Source data 1.** Data for *Figure 8—figure supplement 2*.
DOI: https://doi.org/10.7554/eLife.38852.035

**Source data 1.** Data for *Figure 8*.
DOI: https://doi.org/10.7554/eLife.38852.036

**Figure supplement 1.** Purkinje cell responses to whisker pad stimulation in *Pcp2-Ppp3r1* and *Pcp2-Gria3* mice.
DOI: https://doi.org/10.7554/eLife.38852.032

**Figure supplement 2.** Before induction, touch-induced whisker protraction is not affected by *Pcp2-Ppp3r1* and *Pcp2-Gria3* mutations.
DOI: https://doi.org/10.7554/eLife.38852.034

(difference in simple spike count between WT and *Pcp2-Gria3* PCs during the first 60 ms after air-puff onset: p=0.004; Tukey's post-hoc test after ANOVA (p=0.002, $F$ = 6.681, df = 2), *n* = 35 WT PCs and *n* = 13 *Gria3* KO PCs, next to *n* = 23 *Pcp2-Ppp3r1* KO PCs, all with weak complex spike responses) (*Figure 8D–F*). Thus an independent line of evidence supports the findings made in the *Pcp2-Ppp3r1* mice.

For control we compared the basic electrophysiological profiles of PCs in the three genotypes used in this study. When averaged over the entire period with episodes of stimulation, the overall complex spike rate, simple spike rate and simple spike CV2 value (i.e. parameter for level of irregularity) of PCs in the *Pcp2-Ppp3r1* KO mice were moderately, but significantly, reduced compared to those in WTs (*Figure 8—figure supplement 1A–D*; *Supplementary file 1C*). However, as the *Pcp2-Gria3* mice did not show any significant deviations in these overall firing properties (*Figure 8—figure supplement 1A–D*; *Supplementary file 1C*), it is unlikely that the aberrant firing properties of *Pcp2-Ppp3r1* mice could explain the lack of adaptation at both the behavioral and electrophysiological level. Comparing the response probabilities to whisker pad stimulation we found that both the number of complex spikes and simple spikes after the air-puff were reduced in *Pcp2-Ppp3r1* mice (*Figure 8—figure supplement 1E–J*; *Supplementary file 1C*). The predominantly suppressive simple spike responses were not found in *Pcp2-Gria3* mice, but the latter also had a reduced complex spike response to air-puff stimulation. Since a reduced complex spike response acts permissive for the adaptive increase in the simple spike response, it is unlikely that the observed reduction in complex spike firing would be the cause of the observed lack of simple spike enhancement in both mutant mouse lines. Moreover, the amplitudes of the touch-induced whisker protraction as measured before the induction phase were similar between the WT and the mutant mice (*Figure 8—figure supplement 2*). We therefore conclude that the absence of simple spike potentiation and the concomitant increase in touch-induced whisker protraction is likely due to the absence of parallel fiber LTP caused by the genetic mutations rather than to altered firing patterns of the PCs involved.

## Discussion

In this study, we show for the first time that a brief period of intense sensory stimulation can evoke adaptation of reflexive whisker protraction. Extracellular recordings revealed that the simple spike

activity of the PCs that modulate during whisker movements is congruently increased when the adapted behavior is expressed. These PCs, which are predominantly located in the crus 2 region, show a present but weak complex spike response to whisker stimulation, which appears to act permissive for the occurrence of parallel fiber to PC LTP. This form of plasticity is likely to be one of the main mechanisms underlying this whisker reflex adaptation, as two independent cell-specific mouse models, both of which lack LTP induction at their parallel fiber to PC synapses, did not show any alteration in their whisker protraction or simple spike response following the training protocol with 4 Hz air-puff stimulation. By contrast, the PCs that show a strong complex spike response to whisker stimulation and that are mainly located in crus 1 did not manifest a prominent regulatory role to enhance the simple spike responses or whisker movements in the long-term. Our study highlights how moderate climbing fiber activity may permit induction of PC LTP in a behaviorally relevant context and how this induction may lead to an increase in simple spike modulation when the adapted motor output is expressed.

## Control of whisker movements

Although most mammals have whiskers, only few species use their whiskers to actively explore their environment by making fast, rhythmic whisker movements (*Vincent, 1913*; *Ahl, 1986*; *Welker, 1964*; *Woolsey et al., 1975*). In 'whisking' animals, such as mice and rats, both whisker protraction and retraction are under direct muscle control, while especially whisker retraction can additionally reflect a passive process involving skin elasticity (*Berg and Kleinfeld, 2003*; *Simony et al., 2010*; *Haidarliu et al., 2015*; *Moore et al., 2013*; *Deschênes et al., 2016*). Animals can modify the pattern of whisker movements upon sensory feedback during natural behavior, as has been demonstrated for example during gap crossing and prey capture (*Anjum and Brecht, 2012*; *Voigts et al., 2015*). The neural control of adaptation of reflexive whisker movements is still largely unknown. Given the widespread networks in the brain controlling whisker movements (*Bosman et al., 2011*; *Kleinfeld et al., 1999*), it is likely that multiple brain regions contribute. We show here, at least for a specific reproducible form of whisker adaptation, that parallel fiber to PC LTP and enhancement in PC simple spike activity may contribute to the induction and expression of this form of motor learning, respectively.

## Simple spike firing during normal and adapted whisker movements

Our electrophysiological recordings indicate that the simple spike activity correlates well with whisker protraction, especially in PCs located in crus 2, and that this relation is context-dependent. Under the baseline condition of our paradigm, during the 0.5 Hz whisker pad stimulation, simple spikes correlate positively with the position of the whiskers during protraction on a single-trial basis. The correlation between the rate of simple spikes and that of protraction was also found when comparing the impact of contralateral vs. ipsilateral whisker pad stimulation. The absence of a clear time lag or lead between simple spike activity and whisker movements under this condition suggests that during normal motor performance without sensorimotor mismatch signaling the simple spikes predominantly represent ongoing movement. Our data under baseline conditions are compatible with those obtained by the labs of Chadderton and Léna (*Chen et al., 2016*; *Proville et al., 2014*). In their studies on online motor performance, the simple spike activity of most PCs in the lateral crus 1 and/or crus 2 regions correlated best with protraction of the set point, defined as the slowly varying midpoint between maximal protraction and maximal retraction.

During and after training with 4 Hz air-puff stimulation the temporal dynamics of the simple spikes shifted in that the simple spikes were found to precede the whisker movement and to predict the magnitude of the protraction, suggesting the emergence of an instructive motor signal. Optogenetic stimulation experiments confirmed that increased simple spike firing during the early phase of touch-induced whisker protraction can promote whisker protraction. Thus, the current dataset confirms and expands on previous studies, highlighting a role of the cerebellar PCs injecting additional accelerating and amplifying signals into the cerebellar nuclei during entrainment (*De Zeeuw et al., 1995*).

## Cerebellar plasticity

Synaptic plasticity in the cerebellar cortex has, next to that in the cerebellar and vestibular nuclei (*Lisberger and Miles, 1980*; *Lisberger, 1998*; *Zhang and Linden, 2006*; *McElvain et al., 2010*), generally been recognized as one of the major mechanisms underlying motor learning (*Ito, 2001*; *Ito, 1989*). For forms of motor learning that require a decrease in simple spike activity for expression of the memory, such as eyeblink conditioning (*Halverson et al., 2015*; *ten Brinke et al., 2015*; *Jirenhed et al., 2007*), long-term depression (LTD) of the parallel fiber to PC synapse may play a role during the initial induction stage (*Ito, 1989*; *Koekkoek et al., 2003*). In LTD-deficient mouse models the potential contribution of LTD is most apparent when compensatory mechanisms that involve activation of the molecular layer interneurons are blocked (*Boele et al., 2018*). However, for forms of motor learning that require an increase in simple spike activity for expression of the procedural memory it is less clear which forms of cerebellar cortical plasticity may contribute. Here, we show that increasing whisker protraction by repetitive sensory stimulation requires an increase in simple spike activity and that blocking induction of parallel fiber to PC LTP prevents changes in both spiking and motor activity following the same training paradigm. Possibly, adaptation of the vestibulo-ocular reflex (VOR) follows partly similar learning rules in that various genetic mouse models with impaired induction of parallel fiber to PC LTP show reduced VOR learning (*Gutierrez-Castellanos et al., 2017*; *Rahmati et al., 2014*; *Schonewille et al., 2010*; *Ly et al., 2013*; *Peter et al., 2016*) and that optogenetic stimulation of PCs in the flocculus of the vestibulocerebellum can increase VOR gain (*Voges et al., 2017*). In this respect, it will be interesting to find out to what extent increases in simple spike activity in the flocculus can also be correlated with an entrained increase in VOR gain on a trial-by-trial basis, as we show here for whisker learning.

The differential learning rules highlighted above indicate that different forms of cerebellar plasticity may dominate the induction of different forms of learning (*Hansel et al., 2001*; *Gao et al., 2012*; *D'Angelo et al., 2016*; *De Zeeuw and Ten Brinke, 2015*). The engagement of these rules may depend on the requirements of the downstream circuitries involved (*Suvrathan et al., 2016*; *De Zeeuw and Ten Brinke, 2015*). Indeed, whereas the eyeblink circuitry downstream of the cerebellar nuclei comprises purely excitatory connections and hence requires a simple spike suppression of the inhibitory PCs to mediate closure of the eyelids, the VOR circuitry comprises an additional inhibitory connection and hence requires a simple spike enhancement so as to increase the compensatory eye movement (*De Zeeuw and Ten Brinke, 2015*; *Voges et al., 2017*). The circuitry downstream of the cerebellum that mediates control of whisker movements is complex (*Bosman et al., 2011*). Possibly, the cerebellar nuclei may modulate the trigemino-facial feedback loop in the brainstem that controls the touch-induced whisker protraction (*Bellavance et al., 2017*). This could be done via the intermediate reticular formation, which receives a direct input from the cerebellar nuclei (*Teune et al., 2000*) and projects to the facial nucleus where the whisker motor neurons reside (*Zerari-Mailly et al., 2001*; *Herfst and Brecht, 2008*). As the latter projection is inhibitory (*Deschênes et al., 2016*), the same configuration may hold as described for the VOR pathways (*De Zeeuw and Ten Brinke, 2015*) in that adaptive enhancement of the whisker reflex may require induction of parallel fiber to PC LTP and increases in simple spike activity. Thus, given the current findings and the known neuro-anatomical connections in the brainstem, the picture emerges that cerebellar control of whisker movements follows the general pattern which suggests that the predominant forms of PC plasticity and concomitant changes in simple spike activity align with the requirements downstream in the cerebellar circuitry (*De Zeeuw and Ten Brinke, 2015*).

## Heterogeneous pools of PCs with differential complex spike responses to whisker stimulation

A minority of the PCs we recorded had a high complex spike response probability upon air-puff stimulation of the whisker pad. These PCs were predominantly located in the centro-lateral part of crus 1. Most of the other PCs, in particular those in the medial part of crus 1 and in crus 2, showed a low, yet significant, complex spike response probability to sensory whisker stimulation. In these cells the absence of a strong complex spike response to air-puff stimulation probably acted as a permissive gate to increase the simple spike response following training, which is in line with current theories on cerebellar plasticity (*Coesmans et al., 2004*; *Ito, 2001*; *Lev-Ram et al., 2002*; *Ohtsuki et al., 2009*). The PCs with a relatively high complex spike response probability were not prone for

increases in simple spike activity following our training protocol. Instead, they may dynamically enhance reflexive whisker protraction through increases in their coherent complex spike firing, likely engaging ensemble encoding (*Hoogland et al., 2015*; *Mukamel et al., 2009*; *Ozden et al., 2009*; *Schultz et al., 2009*). This enhancement does not require a repetitive training protocol and also occurs during single trial stimulation. Indeed, these complex spike responses, which tended to precede the active whisker movement, could be correlated to the strength of the touch-induced whisker protraction under baseline conditions. This is in line with previous studies showing that complex spikes can facilitate the initiation of movements and define their amplitude (*Hoogland et al., 2015*; *Kitazawa et al., 1998*; *Welsh et al., 1995*). Thus, PCs with strong complex spike responses to whisker stimulation – especially those located in the D2 zone of crus 1 – show poor simple spike enhancement to mediate whisker adaptation, but they might facilitate execution of touch-induced whisker protraction under baseline conditions by relaying coherent patterns of complex spikes onto the cerebellar nuclei neurons.

## Conclusion

Based on a known form of reflexive whisker movements, we introduced a novel adaptation paradigm and investigated the underlying cerebellar plasticity mechanism and spiking learning rules. A brief period of increased sensory input appeared to be sufficient to induce a lasting impact on touch-induced whisker protraction: the whisker reflex started earlier and had a bigger amplitude. This motor adaptation probably requires induction of parallel fiber LTP in PCs that can be identified by their weak but present complex spike response to sensory stimulation. The resultant increased simple spike firing of these PCs may affect the brainstem loop controlling touch-induced whisker protraction via the reticular formation in the brainstem, in line with optogenetic stimulation experiments. Thus, our study proposes induction of parallel fiber to PC LTP as a cellular mechanism for enhancing PC simple spike responsiveness that facilitates the expression of the entrained whisker protraction.

# Materials and methods

### Key resources table

| Reagent type (species) or resource | Designation | Source or reference | Identifiers | Additional information |
|---|---|---|---|---|
| Strain, strain background (*Mus musculus*) | *Tg(Pcp2-cre)2Mpin; Ppp3r1*$^{tm1Stl}$ | (*Schonewille et al., 2010*) | | C57BL/6J background |
| Strain, strain background (*M. musculus*) | *Tg(Pcp2-cre)2M pin;Gria3*$^{tm2Rsp}$ | (*Gutierrez-Castellanos et al., 2017*) | | C57BL/6J background |
| Strain, strain background (*M. musculus*) | C57BL/6J mice | Charles Rivers | IMSR_JAX:000664 | |
| Strain, strain background (*M. musculus*) | *Tg(Pcp2-cre) 2Mpin; Gt(ROSA)26Sor*$_{tm27.1(CAG-OP4*H134R/tdTomato)Hze}$ | (*Witter et al., 2013*) | | C57BL/6J background |
| Chemical compound, drug | Dextran, Biotin, 3000 MW, Lysine Fixable (BDA-3000) | Thermo Fisher Scientific | D7135 | |
| Chemical compound, drug | Paraform aldehyde | Merck | 1.040005.1000 | |
| chemical compound, drug | Thionine | Sigma | T-3387 | |
| Chemical compound, drug | Gelatin | J.T.Baker | 2124–01 | |

*Continued on next page*

*Continued*

| Reagent type (species) or resource | Designation | Source or reference | Identifiers | Additional information |
|---|---|---|---|---|
| software, algorithm | MATLAB v2012a-v2017a | Mathworks | | |
| Software, algorithm | LabVIEW (for video acquisition) | National Instruments | | |
| Software, algorithm | BWTT Toolbox (for whisker tracking) | http://bwtt.source forge.net; https://github.com /MRIO/BWTT_PP | | |

## Animals

For most of the experiments in this study, we used two different mutant mouse lines on a C57BL/6J background. Comparisons of electrophysiological parameters were always made between the mutant mice and their respective wild-type (WT) littermates, although for easier visualization the WTs were sometimes grouped as indicated in the figure legends. Both mouse lines had been used before and details on their generation have been published. Briefly, *Pcp2-Ppp3r1* mice (*Tg(Pcp2-cre)2Mpin;Ppp3r1^{tm1Stl}*) lacked functional phosphatase 2B (PP2B) specifically in their PCs. They were created by crossing mice in which the gene for the regulatory subunit (Cnbl1) of PP2B was flanked by loxP sites (*Zeng et al., 2001*) with transgenic mice expressing Cre-recombinase under control of the *Pcp2* (*L7*) promoter (*Barski et al., 2000*) as described in *Schonewille et al., 2010*. *Pcp2-cre^{+/-}-Ppp3r1^{f/f}* mice ('*Pcp2-Ppp3r1* mice') were compared with *Pcp2-cre^{-/-}-Ppp3r1^{f/f}* littermate controls. We used 35 WT mice (17 males and 18 females of $21 \pm 9$ weeks of age (average $\pm$s.d.)) and 22 *Pcp2-Ppp3r1* mice (6 males and 16 females of $18 \pm 10$ weeks of age (average $\pm$s.d.)). *Pcp2-Gria3* mice (*Tg(Pcp2-cre)2Mpin;Gria3^{tm2Rsp}*) lacked the AMPA receptor GluA3 subunit specifically in their PCs. They were created by crossing mice in which the *Gria3* gene was flanked by loxP sites (*Sanchis-Segura et al., 2006*) with transgenic mice expressing Cre-recombinase under control of the *Pcp2* promoter (*Barski et al., 2000*) as described in *Gutierrez-Castellanos et al., 2017*. We used *Pcp2-cre^{+/-}-Gria3^{f/f}* mice ('*Pcp2-Gria3* mice') and *Pcp2-cre^{-/-}-Gria3^{f/f}* as littermate controls. We used 5 WT male mice ($25 \pm 3$ weeks of age (average $\pm$s.d.)) and 9 *Pcp2-Gria3* mice (6 males and 3 females of $26 \pm 4$ weeks of age (average $\pm$s.d.)). Mutants and wild-types were measured in random sequence. For the two-photon $Ca^{2+}$ imaging experiments, we used six male C57BL/6J mice (Charles Rivers, Leiden, the Netherlands) of 4–12 weeks of age. The photostimulation experiments were performed on seven mice (3 males and 4 females of $25 \pm 1$ weeks of age (average $\pm$s.d.)) expressing Channelrhodopsin-2 exclusively in their PCs (*Tg(Pcp2-cre)2Mpin;Gt(ROSA)26Sor^{tm27.1(CAG-COP4*H134R/tdTomato)Hze}*) as described previously (*Witter et al., 2013*). All mice were socially housed until surgery and single-housed afterwards. The mice were kept at a 12/12 hr light/dark cycle and had not been used for any invasive procedure (except genotyping shortly after birth) before the start of the experiment. All mice used were specific-pathogen free (SPF). All experimental procedures were approved a priori by an independent animal ethical committee (DEC-Consult, Soest, The Netherlands) as required by Dutch law and conform the relevant institutional regulations of the Erasmus MC and Dutch legislation on animal experimentation. Permissions were obtained under the following license numbers: EMC2656, EMC2933, EMC2998, EMC3001, EMC3168 and AVD101002015273.

## Surgery

All mice that were used for electrophysiology received a magnetic pedestal that was attached to the skull above bregma using Optibond adhesive (Kerr Corporation, Orange, CA) and a craniotomy was made on top of crus 1 and crus 2. The surgical procedures were performed under isoflurane anesthesia (2–4% V/V in $O_2$). Post-surgical pain was treated with 5 mg/kg carprofen ('Rimadyl', Pfizer, New York, NY, USA), 1 µg lidocaine (Braun, Meisingen, Germany), 1 µg bupivacaine (Actavis, Parsippany-Troy Hills, NJ) and 50 µg/kg buprenorphine ('Temgesic', Indivior, Richmond, VA). After 3 days of recovery, mice were habituated to the recording setup during at least two daily sessions of approximately 45 min each. In the recording setup, they were head-fixed using the magnetic pedestal. The mice used for two-photon imaging received a head plate with a sparing on the location of the craniotomy instead of a pedestal. The head plate was attached to the skull with dental cement

(Superbond C and B, Sun Medical Co., Moriyama City, Japan). To prevent the growth of scar tissue, which could affect image quality, two-photon recordings were made on the day of the surgery (recording started at least 1 hr after the termination of anesthesia).

## Whisker stimulation and tracking

Air-puff stimulation to the whisker pad was applied with a frequency of 0.5 Hz s at a distance of approximately 3 mm at an angle of approximately 35° (relative to the body axis). The puffs were delivered using a tube with a diameter of approximately 1 mm with a pressure of ~2 bar and a duration of 30 ms. During the induction period, the stimulation frequency was increased to 4 Hz and 80 puffs were given. In a subset of experiments, a 2 ms air-puff (pre-pulse) was delivered 100 ms prior to the 30 ms puff. Videos of the whiskers were made from above using a bright LED panel as backlight ($\lambda$ = 640 nm) at a frame rate of 1,000 Hz (480 × 500 pixels using an A504k camera from Basler Vision Technologies, Ahrensburg, Germany). The whiskers were not trimmed or cut.

## Electrophysiology

Electrophysiological recordings were performed in awake mice using either glass pipettes (3–6 M$\Omega$) or quartz-coated platinum/tungsten electrodes (2–5 M$\Omega$, outer diameter = 80 µm, Thomas Recording, Giessen, Germany). Unless specified otherwise, recordings were made in *Pcp2-Ppp3r1* WT mice. The latter electrodes were placed in an 8 × 4 matrix (Thomas Recording), with an inter-electrode distance of 305 µm. Prior to the recordings, the mice were lightly anesthetized with isoflurane to remove the dura, bring them in the setup and adjust all manipulators. Recordings started at least 60 min after termination of anesthesia and were made in crus 1 and crus 2 ipsilateral to the side of the whisker pad stimulation at a minimal depth of 500 µm. The electrophysiological signal was digitized at 25 kHz, using a 1–6000 Hz band-pass filter, 22x pre-amplified and stored using a RZ2 multi-channel workstation (Tucker-Davis Technologies, Alachua, FL).

## Neural tracing and electrolytic lesions

For the neural tracing experiments, we used glass electrodes filled with 2 M NaCl for juxtacellular recordings. After a successful recording of a PC, neural tracer was pressure injected (3 × 10 ms with a pressure of 0.7 bar) either from the same pipette re-inserted at the same location or from the second barrel or a double barrel pipette. We used a gold-lectin conjugate has described previously (*Ruigrok et al., 1995*) (*n* = 3) or biotinylated dextran amine (BDA) 3000 (10 mg/ml in 0.9% NaCl; ThermoFisher Scientific, Waltham, MA) (*n* = 7). Five days after the tracer injection, the mice were anesthetized with pentobarbital (80 mg/kg intraperitoneal) and fixated by transcardial perfusion with 4% paraformaldehyde. The brains were removed and sliced (40 µm thick). The slices were processed by Nissl staining. Experiments were included in the analysis if the electrophysiology fulfilled the requirements mentioned below with a recording duration of at least 50 s and if the tracer was clearly visible. For BDA 3000, this implied that it was taken up by the PCs at the injection spot and transported to the axonal boutons a single subgroup in the cerebellar nuclei. BDA 3000 was also found in the inferior olive. For the gold-lectin conjugate the subnucleus of the inferior olive was considered. Based upon the subnuclei of the cerebellar nuclei and/or the inferior olive, the sagittal zone of the recording site was identified according to the scheme published in *Apps and Hawkes, 2009*.

After the recordings made with the quartz/platinum electrodes, electrolytic lesions were applied to selected electrodes in order to retrieve the recording locations. To this end, we applied a DC current of 20 µA for 20 s. This typically resulted in a lesion that could be visualized after Nissl staining of 40 µm thick slices made of perfused brains. We accepted a spot as a true lesion if it was visible in at least two consecutive slices at the same location. In total, we could retrieve 16 successful lesions. Recording locations were approximated using pictures of the entry points of the electrodes in combination with the locations of the lesions.

## Two-photon Ca$^{2+}$ imaging

After the surgery (see above) with the dura mater intact, the surface of the cerebellar cortex was rinsed with extracellular solution composed of (in mM) 150 NaCl, 2.5 KCl, 2 CaCl$_2$, 1 MgCl$_2$ and 10 HEPES (pH 7.4, adjusted with NaOH). After a 30 min recovery period from anesthesia, animals were head-fixed in the recording setup and received a bolus-loading of the cell-permeant fluorescent

$Ca^{2+}$ indicator Cal-520 AM (0.2 mM; AAT Bioquest, Sunnyvale, CA, USA). The dye was first dissolved with 10% w/V Pluronic F-127 in DMSO (Invitrogen) and diluted 20x in the extracellular solution. The dye solution was pressure injected into the molecular layer (50–80 μm below the surface) at 0.35 bar for 5 min. After dye loading, the brain surface was covered with 2% agarose dissolved in saline (0.9% NaCl) in order to reduce motion artefacts and prevent dehydration.

Starting at least 30 min after dye injection, in vivo two-photon $Ca^{2+}$ imaging was performed of the molecular layer using a setup consisting of a titanium sapphire laser (Chameleon Ultra, Coherent, Santa Clara, CA), a TriM Scope II system (LaVisionBioTec, Bielefeld, Germany) mounted on a BX51 microscope with a 20 × 1.0 NA water immersion objective (Olympus, Tokyo, Japan) and GaAsP photomultiplier detectors (Hamamatsu, Iwata City, Japan). A typical recording sampled 40 × 200 μm with a frame rate of approximately 25 Hz.

### Data inclusion
We included all mice measured during this study, with the exception of one mouse where video-analysis revealed that the air-puff was delivered more to the nose than to the whisker pad. Single-unit data was included if the recording was of sufficient quality and reflected the activity of a single PC according to the rules defined below (see section *Electrophysiological analysis*).

### Whisker tracking
Whisker movements were tracked offline as described previously (*Rahmati et al., 2014*) using a method based on the BIOTACT Whisker Tracking Tool (*Perkon et al., 2011*). We used the average angle of all trackable large facial whiskers for further quantification of whisker behavior. The impact of 4 Hz air-puff stimulation on air-puff-triggered whisker movement was quantified using a bootstrap method. First, we took the last 100 trials before induction and divided these randomly in two series of 50. We calculated the differences in whisker position between these two series, and repeated this 1000 times. From this distribution we derived the expected variation after whisker pad air-puff stimulation. We took the 99% confidence interval as the threshold to which we compared the difference between 50 randomly chosen trials after and 50 randomly chosen trials before induction.

### Electrophysiological analysis
Spikes were detected offline using SpikeTrain (Neurasmus, Rotterdam, The Netherlands). A recording was considered to originate from a single PC when it contained both complex spikes (identified by the presence of stereotypic spikelets) and simple spikes, when the minimal inter-spike interval of simple spikes was 3 ms and when each complex spike was followed by a pause in simple spike firing of at least 8 ms. The regularity of simple spike firing was expressed as the local variation (CV2) and calculated as $2|ISI_{n+1}-ISI_n|/(ISI_{n+1}+ISI_n)$ with ISI = inter simple spike interval (*Shin et al., 2007*). Only single-unit recordings of PCs with a minimum recording duration of 200 s were selected for further analysis. However, for the neural tracing experiments (see above), on which no quantitative analysis was performed, we accepted a minimum recording duration of 50 s.

### Two-photon $Ca^{2+}$ imaging analysis
Image analysis was performed offline using custom made software as described and validated previously (*Ozden et al., 2008*; *Ozden et al., 2012*; *De Gruijl et al., 2014*). In short, we performed independent component analysis to define the areas of individual PC dendrites (*Figure 3—figure supplement 2A*). The fluorescent values of all pixels in each region of interest were averaged per frame. These averages were plotted over time using a high-pass filter. A 8% rolling baseline was subtracted with a time window of 0.5 ms (*Ozden et al., 2012*). $Ca^{2+}$ transients were detected using template matching. For the aggregate peri-stimulus time histograms (PSTHs), we calculated per individual frame the number of complex spikes detected and made a PSTH color coding the number of simultaneously detected complex spikes. Based on the total number of complex spikes and dendrites per recording, we calculated the expected number of simultaneous complex spikes per individual frame based upon a Poisson distribution. The actual number of simultaneous complex spikes was compared to this calculated distribution and a p value was derived for each number based upon the Poisson distribution.

## Characterization of sensory responses

For each PC recording, we constructed PSTHs of complex spikes and simple spikes separately using a bin size of 10 ms for display purposes. For further quantitative analyses of the PSTHs, we used a bin size of 1 ms and convolved them with a 21 ms wide Gaussian kernel. Complex spike responses were characterized by their peak amplitude, defined as the maximum of the convolved PSTH and expressed in percentage of trials in which a complex spike occurred within a 1 ms bin. Latencies were taken as the time between stimulus onset and the time of the response peak, as determined from the convolved PSTH. For some analyses, we discriminated between the sensory response period (0–60 ms after stimulus onset) and inter-trial interval (500 to 200 ms before stimulus onset). We considered a PC responsive for sensory stimulation if the peak or trough in the PSTH in the 60 ms after the stimulus onset exceeded the threshold of 3 s.d. above or below the average of the pre-stimulus interval (1 ms bins convolved with a 21 ms Gaussian kernel, pre-stimulus interval 200 ms before stimulus onset). Long-term stability of electrophysiological recordings was verified by heat maps of time-shifted PSTHs. The time-shifted PSTH was processed by calculating the simple spike PSTH for 20 air-puffs per row, which were shifted by five air-puffs between neighboring rows. The simple spike rates per row are calculated at 1 ms resolution and convolved with a 21 ms Gaussian kernel and color-coded relative to baseline firing rate (−1000 to −200 ms relative to air-puff time).

## Cluster analysis

A principal component analysis showed that the heterogeneity among the sensory complex spike responses was driven almost exclusively by one parameter, the maximum amplitude peak of the convolved complex spike PSTH. We performed a univariate Gaussian mixture model using only that variable. The Bayesian information criterion (BIC) indicated that the model with two components with unequal variances yielded the best approximation of the data. Then we applied the function Mclust (data) in R (R Foundation, Vienna, Austria) which use the expectation-maximization algorithm in order to assert the main parameters of the resulting models (probability, mean and variance of each population).

## Spike-whisker movement correlation matrix

Trial-by-trial correlation between instantaneous simple spike firing rate and whisker position was performed as described before (*ten Brinke et al., 2015*). In short: spike density functions were computed for all trials by convolving spike occurrences across 1 ms bins with an 8 ms Gaussian kernel. Both spike and whisker data were aligned to the 200 ms baseline. For cell groups, data was standardized for each cell for each correlation, and then pooled. The spike-whisker Pearson correlation coefficient R was calculated in bins of 10 ms, resulting in a 40 × 40 R-value matrix showing correlations for −100 to 300 ms around the air-puff presentation.

## Statistical analysis

Group sizes of the blindly acquired data sets were not defined a priori as the effect size and variation were not known beforehand. A post-hoc power calculation based upon the results of the potentiation of the PC responses to whisker pad stimulation of the 'weak complex spike responders' indicated a minimum group size of 12 PCs ($\alpha$ = 5%, $\beta$ = 20%, $\Delta$ = 9.65%, s.d. = 10.59%, paired *t* test). This number was obtained for the 'weak complex spike responders' in WT (*n* = 35), *Pcp2-Ppp3r1* (*n* = 21) and *Pcp2-Gria3* PCs (*n* = 13), as well as for the relatively rare 'strong complex spike responders' in WT mice (*n* = 20). This was further substantiated by other independent analyses, including ANOVA and linear regression, as described in the Results section. Variations in success rate, especially considering recordings of longer duration in combination with video tracking, explain why some groups are larger than others. Data was excluded only in case of a signal to noise ratio that was insufficient to warrant reliable analysis. For data visualization and statistical analysis, we counted the number of PCs as the number of replicates for the spike-based analyses and the number of mice for the behavior-based analyses. We tested whether the observed increase in coherence after sensory stimulation (*Figure 3—figure supplement 2D–G*) was more than expected from the increased firing rate induced by the stimulation. The expected coherence based on the firing rate was calculated from 1000 bootstrapped traces from the inhomogeneous Poisson spike trains made for each neuron. The resultant distribution was compared to the measured distribution using a two-sample

Kolmogorov-Smirnov test. Stacked line plots were generated by cumulating the values of all subjects per time point. Thus, the first line (darkest color) represents the first subject, the second line the sum of the first two, the third line the sum of the first three, etcetera. The data were divided by the number of subjects, so that the last line (brightest color) represents, next to the increase from the one but last value, also the population average. Sample size and measures for mean and variation are specified throughout the text and figure legends. For normally distributed data (as evaluated using the Kolmogorov-Smirnov test) parametric tests were used. Comparisons were always made with 2-sided tests when applicable. Unpaired *t* test were always made with Welch correction for possible differences in s.d..

## Data and software availability

The data for the box plots is available as Source Data Files. Custom written Matlab code to complement the whisker tracking analysis by the BIOTACT Whisker Tracking Tool was used as described previously (*Rahmati et al., 2014*) and is available via GitHub (*Spanke and Negrello, 2018*; copy archived at https://github.com/elifesciences-publications/BWTT_PP).

## Acknowledgements

Financial support was provided by the Netherlands Organization for Scientific Research (NWO-ALW; CIDZ), the Dutch Organization for Medical Sciences (ZonMW; CIDZ), Life Sciences (CIDZ), ERC-adv and ERC-POC (CIDZ) and the China Scholarship Council (No. 2010623033; CJ). We thank E Haasdijk, E Goedknegt, M Rutteman, and ACHG IJpelaar for technical assistance and Drs. TJH Ruigrok, SKE Koekkoek, JJ White and M Schonewille of the Neuroscience Department at the Erasmus Medical Center for their input and scientific discussions, as well as Dr. D Rizopoulos of the Biostatistics Department at the Erasmus Medical Center and Dr. D Vidotto from Tilburg University for statistical consulting regarding the cluster analysis.

## Additional information

### Funding

| Funder | Grant reference number | Author |
| --- | --- | --- |
| Nederlandse Organisatie voor Wetenschappelijk Onderzoek | ALW | Chris I De Zeeuw |
| ZonMw | | Chris I De Zeeuw |
| European Research Council | ERC-Advanced Grant | Chris I De Zeeuw |
| European Research Council | ERC-PoC | Chris I De Zeeuw |
| China Scholarship Council | 2010623033 | Chiheng Ju |

The funders had no role in study design, data collection and interpretation, or the decision to submit the work for publication.

### Author contributions

Vincenzo Romano, Laurens WJ Bosman, Conceptualization, Resources, Data curation, Software, Formal analysis, Supervision, Validation, Investigation, Visualization, Methodology, Writing—original draft, Project administration, Writing—review and editing; Licia De Propris, Conceptualization, Formal analysis, Validation, Investigation, Visualization, Methodology, Writing—original draft; Pascal Warnaar, Software, Formal analysis, Validation; Michiel M ten Brinke, Software, Formal analysis, Validation, Methodology; Sander Lindeman, Software, Formal analysis, Validation, Investigation, Visualization; Chiheng Ju, Funding acquisition, Investigation, Methodology; Arthiha Velauthapillai, Software, Formal analysis, Methodology; Jochen K Spanke, Software, Formal analysis, Visualization; Emily Middendorp Guerra, Software, Formal analysis; Tycho M Hoogland, Resources, Data curation, Methodology; Mario Negrello, Resources, Data curation, Software, Formal analysis, Supervision, Validation, Visualization, Methodology; Egidio D'Angelo, Conceptualization, Methodology; Chris I De

Zeeuw, Conceptualization, Resources, Supervision, Funding acquisition, Validation, Visualization, Methodology, Project administration, Writing—review and editing

### Author ORCIDs
Vincenzo Romano (iD) http://orcid.org/0000-0002-4449-6541
Laurens WJ Bosman (iD) http://orcid.org/0000-0001-9497-0566
Michiel M ten Brinke (iD) https://orcid.org/0000-0002-9478-1586
Chris I De Zeeuw (iD) http://orcid.org/0000-0001-5628-8187

### Ethics
Animal experimentation: All experimental procedures were approved a priori by an independent animal ethical committee (DEC-Consult, Soest, The Netherlands) as required by Dutch law and conform the relevant institutional regulations of the Erasmus MC and Dutch legislation on animal experimentation. Permissions were obtained under the following license numbers: EMC2656, EMC2933, EMC2998, EMC3001, EMC3168 and AVD101002015273.

### Decision letter and Author response
Decision letter https://doi.org/10.7554/eLife.38852.040
Author response https://doi.org/10.7554/eLife.38852.041

## Additional files
### Supplementary files
• Supplementary file 1 This file contains tables with statistical evaluations of data represented in some of the figures. (**A**) Overview of statistical tests on whisker movements – belonging to *Figure 1—figure supplement 2*. (**B**) Overview of statistical tests – belonging to *Figure 5*. (**C**) Overview of statistical tests – belonging to *Figure 8—figure supplement 1*.
DOI: https://doi.org/10.7554/eLife.38852.037

• Transparent reporting form
DOI: https://doi.org/10.7554/eLife.38852.038

### Data availability
Source data files for all box plots are provided for Figures 3, 5 and 8 and for Figure 1—figure supplement 2, Figure 5—figure supplements 1, 2, 3 and 4, and Figure 8—figure supplements 1 and 2. Custom written Matlab code to complement the whisker tracking analysis by the BIOTACT Whisker Tracking Tool was used as described previously (Rahmati et al., 2014) and is available via GitHub (https://github.com/MRIO/BWTT_PP; copy archived at https://github.com/elifesciences-publications/BWTT_PP).

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
