## [Decision Letter]

[Editors’ note: this article was originally rejected after discussions between the reviewers, but the authors were invited to resubmit after an appeal against the decision.]

Thank you for submitting your work entitled "Adaptation of Whisker Movements Requires Cerebellar Potentiation" for consideration by *eLife*. Your paper has been reviewed by two peer reviewers and the evaluation has been overseen by a Reviewing Editor and a Senior Editor. The paper contains novel and seemingly exciting data. Yet reviewers find that aspects of the paper appear incomplete in terms of both the depth of the data and the interpretation of the data. All readers note difficulty is establishing the major conclusions of the work. After some consideration, are afraid that we cannot publish this paper.

Some key issues:

The claim of a reflexive response upon puffs to the whiskers (Figure 2) is known (Bellavance et al., 2017), much as the present kinetic data is more complete.

The claim of amplitude control is a bit strong. Figure 5A shows a single, fractional increment, not a monotonic curve.

The evidence for a solely cerebellar contribution to the plasticity of amplitude control in whisking seems overstated. In L7-PP2B mutants the block of the behavioral changes is incomplete while the data from GluA3-deficient mice is difficult to interpret; reviewer one details this.

Please note that the observation that only a fraction of cells change their behavior is seen in a positive light and is appreciated; there is no reason for neighboring cells to have similar behavior.

We apologize that we cannot be positive at this time.

*Reviewer #1:*

The paper by Romano et al. explores the relationship between whisker movements and Purkinje cell (PC) spiking responses in the lateral hemispheres of the mouse cerebellum. This is an interesting and important area of research that is poorly understood. The authors use an air puff to evoke reflexive movements and consequently apply electrical stimulation to produce sustained changes in the nature of these reflexes. In addition to changing evoked whisker movements, this stimulation paradigm is proposed to induce a form of long term plasticity at the parallel fibre-Purkinje cell synapse. Although the topic is interesting, the paper presents a complicated, convoluted picture and the main results are puzzling.

Major comments:

1) A general weakness of the study is the stimulation paradigm (4Hz tetanic stimulation of the whisker pad) which is not ethologically relevant. Unlike say prism glasses in smooth pursuit, or VOR turntables, it is difficult to interpret how the system should respond. This in turn makes it difficult to interpret the functional relevance of the data.

2) Although the electrical stimulation does seem to change the pattern of evoked whisking, the evidence that this is solely a cerebellar phenomenon is not strong. The authors use two transgenic mouse models to argue their point, but the data is not at all conclusive: in L7-PP2B mutants the block of the behavioural changes is incomplete (subsection “Expression of PP2B in Purkinje cells is required for increased protraction and simple spike firing following 4 Hz tetanic stimulation”) and is also associated with aberrant ongoing cerebellar activity (subsection “Basic PC spiking responses in L7-PP2B and L7-GluA3 mice”). The data from GluA3-deficient mice is difficult to interpret as it is described differently, but here abnormalities in complex spiking are also reported. It is therefore quite possible that ongoing cerebellar dysfunction in these lines contributes increased 'noise' to brain-wide sensorimotor processing, but that cerebellum does not provide the locus for adaptive changes. In the brains of wild-type mice, adaptation could occur at many different loci in the periphery and CNS. Despite these uncertainties, the authors assert a causal relationship between PC activity and the amplitude of whisker movements throughout the paper. This claim is first made in the Abstract, but the evidence for this is also not strong. The evidence as presented – from optical stimulation L7-Ai27 mice – shows that increased activity in PCs can influence movement, but this doesn't rule out that modulations of whisker amplitude and simple spike (SS) rate are covariant and driven via alternative site(s) in the brain under normal conditions.

3) It is important to establish whether the relationship between whisker kinematics and single cell PC activity really changing following stimulation. The authors could recover the linear relationship between whisker position and PC SS rate and test whether changes in the slope of this relationship are altered pre- and post-stimulation. It should also be possible to recover the cross-correlation between SS rate and whisker position and examine this relationship for all individual cells pre- and post- stimulation. Currently a smoothed group mean is presented in Figure 8F, but this doesn't reveal how individual cells with different temporal relationships (e.g. leading and lagging PCs) alter their responses. Previously, a range of temporal relationships have been reported between changes in whisker movement and SS rate. It is necessary to address this heterogeneity directly in order to make the current results interpretable.

4) Ultimately the authors show that the relationship between SS and whisker movement is changed in some, but not all, PCs following tetanic electrical stimulation of the whisker pad. The result doesn't make much sense in the context of the previous literature on cerebellar learning, which predicts that the relationship between intention and action are be rapidly learnt and relearnt. The authors appear to be suggesting that this relationship can only be adjusted in one of two lobules in cerebellum – such a result is extremely surprising and requires closer attention – for instance comparing the changes in kinematic tuning of individual PCs before and after stimulation in both lobules.

*Reviewer #2:*

Using whisking behavior, Romano and colleagues have used the cerebellum as model for understanding how features of movement behaviors are encoded and controlled by neural circuits. They demonstrate that Purkinje cells not only respond to whisker touch and encode the position of a whicker during its movement, but they also actively control the amplitude of the whisker responses. Using genetic manipulations that the group has used before, they also demonstrate that short- and long-term modification of whisker behavior and the accompanying spike properties are perturbed in mice with LTP deficits.

This is a very interesting and solid manuscript. The text is well written and the rationale for why the specific experiments were chosen is, for the most part, well justified. The data panels are beautiful and the expertise of the group is apparent in the elegance of the data sets. However, in a few minor instances I have a number of questions/comments for the authors, mainly regarding the interpretation of the data and clarity of some explanations.

1) One of the major findings of the paper is that cerebellar crus 1 and 2 have Purkinje cells that have either strong or weak responses to the whisker stimulations. It was not so clear to me whether these lobules also contain cells with no responses at all, and if yes, what was their distribution? Also, it would be nice to see what cells in other non-responsive lobules look like. It just seems that rather than a binary code of strong versus weak responses, there could be a spectrum of responses. If not, this should be demonstrated more clearly (or at the very least discussed).

2) In subsection “Coherent complex spike firing is specifically enhanced by whisker pad stimulation” the authors state "…and this changed towards the involvement of many PCs…" What exactly do you mean by this? How many is "many"? Was there a change in the overall distribution, ie I am trying to get a sense of whether this change reflects some kind of logical recruitment of PCs in some kind of predictable pattern. Was the spatial and temporal change consistent across mice or was the change perhaps less predictable and purely an impact of the number of PCs that were involved in any given animal?

3) The authors have a section regarding "Contralateral whicker pad stimulation…" What is the anatomical basis for these effects? I am also confused about what the background rationale was for these experiments.

4) The optogenetics experiments were performed with 100ms pulses of light. I very much like this experiment, but the details for why this specific paradigm was chosen is not clear. How does this 100ms timing impact the PCs responses over time? Is the strength of the responses reliably strong to make the conclusions that you have made? What about feedback effects? The authors have used this paradigm to test whether the PCs actually modulate the whisker movements but it is not clear whether a clear-cut conclusion about causative effects can be made. Perhaps it can, but the authors need to do a better job of fully explaining all the details of this experiment, including all the possible scenarios that might be occurring upon stimulation.

5) The authors report that the 4Hz stimulation was sufficient to induce an increase in maximal protraction. The data are interesting, but what is the impact on the different circuits that control whisker behavior? It is not clear what global impact this manipulation has on the brain. I suppose I am having a bit of troubling linking this gross manipulation to the very specific spike data that was presented. This goes back to my previous query about anatomical pathways.

6) In the third paragraph of subsection “Tetanic 4 Hz stimulation leads to acceleration of the simple spike response and to stronger protraction of the whiskers”, the authors report that complex spike and simple spike responses to each air puff were weakened…. However, it is not so clear to me why this is the case. Was this result predicted? If yes (or no), what was the rationale?

7) In subsection “Tetanic 4 Hz stimulation specifically potentiates simple spike firing in PCs with weak complex spike responses”, what is the authors definition of a "strong" complex spike responses? Are you talking about frequency, probability, amplitude, duration?

8) I do not understand why 3 out of the 13 L7-PP2B mice did show touch evoked whisker protractions. Being a genetic experiment, there must be some explanation for the presence of responses in these 3 animals.

[Editors’ note: what now follows is the decision letter after the authors submitted for further consideration.]

Thank you for submitting your article "Potentiation of cerebellar Purkinje cells facilitates whisker reflex adaptation through increased simple spike activity" for consideration by *eLife*. Your article and rebuttal letter have been reviewed by Reviewing Editor David Kleinfeld and Eve Marder as the Senior Editor.

We are pleased to consider acceptance of your paper for publication pending one correction. The statement "… whisker retraction is largely mediated by a passive process.…" is an overstatement. Retraction can be active – as pointed out in the manuscript "Rhythmic whisking in rat: Retraction as well as protraction is under active muscular control" by Berg et al., 2003, as well as in Hill et al. (J Neurosci 2008) and Bellavance et al., 2017. Retraction occurs through the action of the maxolabialis and nasolabialis extrinsic retractor muscles. These muscles are driven by input from the preBotzinger complex (Moore et al., 2013; Deschenes et al., 2017) as well as from other sources. Thus the retraction seen in Figure 1C,E can in principle be an active process. Please correct the text and remove the call-out "passive" from Figure 1.

Beyond this, we congratulate you on very nice work.

Romano and colleagues provide evidence that changes in the reflexive motion of the vibrissae in rodent emerges from potentiation at parallel fiber to Purkinje cell synapses in a manner consistent with accepted rules for cerebellar plasticity.

---

## [Author Response]

[Editors’ note: the author responses to the first round of peer review follow.]

We would like to thank you and the reviewers for the constructive comments that have helped us to improve our manuscript. We appreciate it that you found our data to be novel and exciting, and allowed us to resubmit following the appeal. As detailed below, we have addressed all comments of both you and the reviewers. We have re-written large parts of the manuscript, avoiding phrases that might have created confusion. In order to make it easier to follow our main line of reasoning, we have also moved some of the figures to the supplementary material. We feel that this makes the manuscript more focused, while keeping all relevant details available. In line with the suggestions of the reviewers, we have modified some figures and added a number of extra supplementary figures with new data analyses further strengthening the conclusions of our study. Finally, we have changed the title of our manuscript into “Potentiation of cerebellar Purkinje cells facilitates whisker reflex adaptation through increased simple spike activity”.

Some key issues:The claim of a reflexive response upon puffs to the whiskers (Figure 2) is known (Bellavance et al., 2017), much as the present kinetic data is more complete.

It is correct that it is known that sensory input can trigger whisker responses (e.g., Ferezou et al., 2007; Bellavance et al., 2017; Brown and Raman, 2018), but none of these studies described a form of adaptation of this reflexive behavior, let alone the plasticity mechanisms that underlie it. We start off by describing the reflex itself before we go into the adaptation paradigm, because we want to compare the outcome of the motor learning paradigm to that of baseline motor performance. Moreover, we have to make sure that our measurements are reproducible. Still, to comply to the comments of reviewer 1 we simplified Figure 1 and made Figure 2 a supplementary figure. In the text, we improved the description of previous work and put more emphasis on the novelty of the reflex adaptation mechanism we present in the current manuscript.

The claim of amplitude control is a bit strong. Figure 5A shows a single, fractional increment, not a monotonic curve.

In Figure 5 (now Figure 3) we present evidence that the reflexive whisker protraction is stronger in trials with a complex spike than those without. Complex spikes are all-or-none events occurring at a low frequency: either there is one complex spike or there is none. Occurrence of multiple complex spikes during the initial response window is rare. For this reason the data cannot be presented as a monotonic curve and they have to be presented as a single fractional increment. Even though we cannot present a monotonic curve based on the number of complex spikes observed per trial, we added more quantification of the potential impact of complex spikes (Figure 3—figure supplement 1). We expect that this new analysis yields a better understanding of the putative impact of complex spike firing on reflexive whisker movements. Despite these additional data as well as despite the fact that our results are in line with previous studies showing that complex spikes can facilitate the initiation of movements and influence their amplitude (Welsh et al., 1995; Kitazawa et al., 1998; Hoogland et al., 2015; Ten Brinke et al., 2015), we have toned down our claim that Purkinje cells with a high complex spike response probability determine reflexive whisker protraction.

The evidence for a solely cerebellar contribution to the plasticity of amplitude control in whisking seems overstated. In L7-PP2B mutants the block of the behavioral changes is incomplete while the data from GluA3-deficient mice is difficult to interpret; reviewer one details this.

Obviously, neuronal control of movement is a network phenomenon involving many brain regions (Bosman et al., 2011). However, we present four lines of evidence that the cerebellum provides at least one of the main forms of plasticity that contributes to this form of learning. First, we demonstrate a correlation between whisker movement and cerebellar Purkinje cell activity before, during and after the induction phase. In the latter, changes in the instantaneous simple spike rate precede the amplitude of the whisker protraction by about 20 ms. We noticed that the increased simple spike firing after induction depended – as cerebellar theory predicts, but has not been shown experimentally in a behaviorally relevant context – on the relatively low occurrence of complex spikes. Second, our experiments with optogenetic approaches (executed in a cell-specific fashion in the cerebellum), show that the increased simple spike firing can indeed affect whisker protraction. Third, we show that in L7-PP2B mice (lacking LTP specifically in the Purkinje cells) the maximal protraction goes from 8.6 ± 3.9° before to 8.7 ± 4.4° (avg ± sd; n = 13 mice) after induction. We consider this as an absence of a significant change (p = 0.647; paired t-test). The outcomes of the statistical tests argue against a residual response in L7-PP2B mice. We do agree with reviewer 1 though that a genetic mutation can induce compensatory mechanisms (in fact, it probably always does), and therefore we included a second, independent mutant mouse line as the fourth line of evidence. In the Purkinje cell specific L7-GluA3 mutants we essentially reproduced the L7-PP2B phenotype. We acknowledge the occurrence of compensatory mechanisms and in fact make these explicit in Figure 11–figure supplement 1 (now: Figure 8—figure supplement 1). Therefore, we agree that each of the 4 lines of evidence is by itself not sufficient to convincingly show that the cerebellum is the sole site of plasticity, but given the synergy of our different experiments, we feel that we have a strong case in stating that the cerebellum provides at least an essential contribution. To clarify the misunderstanding that L7-PP2B mice could have a residual adaptation of the whisker movement we added a new panel to Figure 7 (Figure 10 in the previous version). In Figure 7E we show the max protraction amplitude for WT and PP2B before and after the induction protocol. The former pie charts, which are less informative than the quantitative analyses shown in the revised manuscript, have been removed to prevent further confusion.

Please note that the observation that only a fraction of cells change their behavior is seen in a positive light and is appreciated; there is no reason for neighboring cells to have similar behavior.We apologize that we cannot be positive at this time.

Reviewer #1:

The paper by Romano et al. explores the relationship between whisker movements and Purkinje cell (PC) spiking responses in the lateral hemispheres of the mouse cerebellum. This is an interesting and important area of research that is poorly understood. The authors use an air puff to evoke reflexive movements and consequently apply electrical stimulation to produce sustained changes in the nature of these reflexes. In addition to changing evoked whisker movements, this stimulation paradigm is proposed to induce a form of long term plasticity at the parallel fibre-Purkinje cell synapse. Although the topic is interesting, the paper presents a complicated, convoluted picture and the main results are puzzling.

Reviewer 1 talks about “electrical stimulation” in this summary statement as well as in his major comments 2 and 4. This is quite remarkable, as we did NOT apply electrical stimulation, but relevant natural sensory stimulation. This is mentioned at all parts of the manuscript – see for instance Figure 8A (now: Figure 5A) that mentions “80 puffs at 4 Hz”. We do not understand how this misunderstanding came about. The most likely explanation is that the use of the word "tetanic" might have been associated with electrical stimulation. To avoid this potential confusion we took out the word "tetanic" and we now consistently refer to the induction protocol as "4 Hz air-puff stimulation".

Major comments:1) A general weakness of the study is the stimulation paradigm (4Hz tetanic stimulation of the whisker pad) which is not ethologically relevant. Unlike say prism glasses in smooth pursuit, or VOR turntables, it is difficult to interpret how the system should respond. This in turn makes it difficult to interpret the functional relevance of the data.

The reviewer questions the ethological relevance of our tetanic stimulation paradigm. In case the reviewer refers to their notion that we used electrical stimulation, please see above and note that this was not the case; we only used sensory stimulation of the natural sensory whisker organs with air-puffs. In case the reviewer also finds the 4 Hz air-puff stimulation ethologically irrelevant, we respectfully disagree. To date, during active exploration mice repeatedly touch objects (as for instance well described during gap crossing; Voigts et al., 2015). Likewise, a similar phenomenon occurs during running through a tunnel-like structure, which is common for mice. Brief periods of intensive touch are therefore behaviorally relevant for mice. Object touch triggers reflexive protraction (e.g., see Bellavance et al., 2017), providing an ethological context of our data. We therefore are confident that our paradigm is not only as ethologically relevant as most other behavioral paradigms commonly used in neuroscience, but also constitutes a simple and powerful tool to address many physiological research questions.

2) Although the electrical stimulation does seem to change the pattern of evoked whisking, the evidence that this is solely a cerebellar phenomenon is not strong. The authors use two transgenic mouse models to argue their point, but the data is not at all conclusive: in L7-PP2B mutants the block of the behavioural changes is incomplete (subsection “Expression of PP2B in Purkinje cells is required for increased protraction and simple spike firing following 4 Hz tetanic stimulation”) and is also associated with aberrant ongoing cerebellar activity (subsection “Basic PC spiking responses in L7-PP2B and L7-GluA3 mice”). The data from GluA3-deficient mice is difficult to interpret as it is described differently, but here abnormalities in complex spiking are also reported. It is therefore quite possible that ongoing cerebellar dysfunction in these lines contributes increased 'noise' to brain-wide sensorimotor processing, but that cerebellum does not provide the locus for adaptive changes. In the brains of wild-type mice, adaptation could occur at many different loci in the periphery and CNS. Despite these uncertainties, the authors assert a causal relationship between PC activity and the amplitude of whisker movements throughout the paper. This claim is first made in the Abstract, but the evidence for this is also not strong. The evidence as presented – from optical stimulation L7-Ai27 mice – shows that increased activity in PCs can influence movement, but this doesn't rule out that modulations of whisker amplitude and simple spike (SS) rate are covariant and driven via alternative site(s) in the brain under normal conditions.

The second major comment of reviewer 1 is reflected in the third Key Issue of the Reviewing Editor (see above). We realise that the previous way of presenting our results has created confusion about possible residual effects on motor adaptation in our mutant mouse lines. We have adapted Figure 7 to better represent the variability in our data. We also addressed this issue more explicitly in our statistical analysis and this further confirms that the perceived residual effect in the mutant mice falls well within the normal variation. This implies that one can conclude that we have been unable to detect any significant sign of adaptation in the cerebellar mutants. Furthermore, we have included a new supplementary figure (Figure 8—figure supplement 2), in which we address the maximum amplitude of the baseline reflexive touch-induced whisker protraction (before applying the training stimuli). We found that there is no significant difference between wild type and mutant mice (p = 0.860; ANOVA). This indicates that the amplitude of the unperturbed whisker reflex is not affected by the changes in Purkinje cell firing described in Figure 8—figure supplement 1. These findings, together with the results of the Purkinje cell electrophysiology and the optogenetic stimulation of Purkinje cells, suggest that parallel fiber LTP contributes to whisker reflex adaptation. Still, we agree with the reviewer that modulations of whisker amplitude may also be partly driven via alternative site(s) in the brain. We have now highlighted this more explicitly in the Discussion.

3) It is important to establish whether the relationship between whisker kinematics and single cell PC activity really changing following stimulation. The authors could recover the linear relationship between whisker position and PC SS rate and test whether changes in the slope of this relationship are altered pre- and post-stimulation. It should also be possible to recover the cross-correlation between SS rate and whisker position and examine this relationship for all individual cells pre- and post- stimulation. Currently a smoothed group mean is presented in Figure 8F, but this doesn't reveal how individual cells with different temporal relationships (e.g. leading and lagging PCs) alter their responses. Previously, a range of temporal relationships have been reported between changes in whisker movement and SS rate. It is necessary to address this heterogeneity directly in order to make the current results interpretable.

We appreciate the observation of reviewer 1 on the heterogeneity of Purkinje cell activity and its relation with behavior. Our analyses were already based on individual cells, but we have included some extra panels making the impact of individual Purkinje cells more explicit. We have included a more detailed analysis in Figure 5F (the topic of former Figure 8 is now presented in Figure 5), showing – for each Purkinje cell – the shift in the moment of maximal correlation before and after induction. Moreover, we implemented also the suggestion of testing whether changes in the slope of the spike/whisker relationship were altered pre- and post-stimulation (Figure 5—figure supplement 1C). This analysis revealed that after the 4 Hz stimulation, it is not the relation between simple spikes and whisker position that changes, but it is the number and the timing of the simple spikes that makes the difference. We thank the reviewer for suggesting these useful additions.

4) Ultimately the authors show that the relationship between SS and whisker movement is changed in some, but not all, PCs following tetanic electrical stimulation of the whisker pad. The result doesn't make much sense in the context of the previous literature on cerebellar learning, which predicts that the relationship between intention and action are be rapidly learnt and relearnt. The authors appear to be suggesting that this relationship can only be adjusted in one of two lobules in cerebellum – such a result is extremely surprising and requires closer attention – for instance comparing the changes in kinematic tuning of individual PCs before and after stimulation in both lobules.

The reviewer rightfully refers to the heterogeneity found in our dataset. However, this is not contradictory to cerebellar theories. There is not a single cerebellar theory that states that all Purkinje cells, or all lobules / modules / zones, should change during a specific form of motor learning. In contrast, it has never been found for any behavioral paradigm in thousands of cerebellar studies and it is clear now that anatomical and functional heterogeneity among cerebellar regions is ubiquitous. However, we agree that in some parts of the text (mainly in the Abstract), we were too strict in defining the regions in which specific phenomena were observed. It is not the case that crus 1 does one thing and crus 2 another without any relation between these two. To clarify this, we have adapted our phrasings, giving attention to the observed gradients in activity. We have also added more maps (Figure 3—figure supplement 1) as well as comparative quantitative data (Figure 5—figure supplement 1) to further illustrate the spatial heterogeneity, including that of crus 1 and crus 2. Overall, we thank the reviewer for pointing out this problem and we are confident that the new description of spatial heterogeneity – which we found to be abundant – is more in line with our data.

Reviewer #2:

Using whisking behavior, Romano and colleagues have used the cerebellum as model for understanding how features of movement behaviors are encoded and controlled by neural circuits. They demonstrate that Purkinje cells not only respond to whisker touch and encode the position of a whicker during its movement, but they also actively control the amplitude of the whisker responses. Using genetic manipulations that the group has used before, they also demonstrate that short- and long-term modification of whisker behavior and the accompanying spike properties are perturbed in mice with LTP deficits.This is a very interesting and solid manuscript. The text is well written and the rationale for why the specific experiments were chosen is, for the most part, well justified. The data panels are beautiful and the expertise of the group is apparent in the elegance of the data sets. However, in a few minor instances I have a number of questions/comments for the authors, mainly regarding the interpretation of the data and clarity of some explanations.

We thank reviewer 2 for these constructive remarks. They were, like those of reviewer 1, instrumental in drafting an improved version of the manuscript. We thank reviewer 2 also for the encouraging positive remarks on the clarity of the text and figures and the underlying data.

1) One of the major findings of the paper is that cerebellar crus 1 and 2 have Purkinje cells that have either strong or weak responses to the whisker stimulations. It was not so clear to me whether these lobules also contain cells with no responses at all, and if yes, what was their distribution? Also, it would be nice to see what cells in other non-responsive lobules look like.

We clarify in the new text that 2 out of the 132 cells show no significant modulation of their complex spikes nor that of simple spikes. The seminal work of Welker’s lab has shown that Purkinje cells that respond to whisker stimulation are widely distributed over the cerebellar cortex (e.g., see Shambes et al., Brain Behav Evol, 1978, and Joseph et al., Brain Behav Evol, 1978). However, most contemporary studies focussed on crus 1 and crus 2, and so did we. We fully agree that a wider sampling area can be of great interest – especially in view of the heterogeneity we found within individual lobules. This would, however, require a serious expansion of our dataset and is therefore unfortunately outside the scope of this study.

It just seems that rather than a binary code of strong versus weak responses, there could be a spectrum of responses. If not, this should be demonstrated more clearly (or at the very least discussed).

The reviewer refers to a discussion that we have had often in our lab. In an attempt to end this discussion – clusters or continuum? – we consulted a biostatistician. The BIC analysis indicated that it was most appropriate to consider our dataset as two separate groups. As we ourselves were also only partially convinced, we already presented most analyses on the complete dataset. In the revision, we have expanded this further and also – following the reviewer’s suggestion – made the restricted value of the clustering more clear in the new version of the manuscript, highlighting the predefined statistical criteria.

2) In subsection “Coherent complex spike firing is specifically enhanced by whisker pad stimulation” the authors state "…and this changed towards the involvement of many PCs…" What exactly do you mean by this? How many is "many"? Was there a change in the overall distribution, ie I am trying to get a sense of whether this change reflects some kind of logical recruitment of PCs in some kind of predictable pattern. Was the spatial and temporal change consistent across mice or was the change perhaps less predictable and purely an impact of the number of PCs that were involved in any given animal?

We agree with the reviewer that these are important aspects of Purkinje cell encoding. We have adapted the text to better describe the distribution of co-active Purkinje cells. An analysis across mice is presented in Figure 3—figure supplement 2G. Further analysis of complex spike coherence in relation to sensory stimulation is interesting indeed, but we feel it would drift away too far from the main topic of this study and we prefer to make it subject of another paper.

3) The authors have a section regarding "Contralateral whicker pad stimulation…" What is the anatomical basis for these effects? I am also confused about what the background rationale was for these experiments.

We thank the reviewer for pointing out this apparent lack of clarity in the previous version. The rationale for these analyses originates from the observation described in Figure 1—figure supplement 2 that contralateral stimulation leads to stronger whisker protraction. We reasoned that if simple spikes positively correlate with whisker movement, then contralateral stimulation should involve stronger simple spike responses. This indeed turned out to be the case. As now mentioned in the manuscript, many connections in the whisker system are bilateral, including those of the mossy fiber pathways (e.g. Bosman et al., 2011), providing many opportunities for cross-talk between the hemispheres.

4) The optogenetics experiments were performed with 100ms pulses of light. I very much like this experiment, but the details for why this specific paradigm was chosen is not clear. How does this 100ms timing impact the PCs responses over time? Is the strength of the responses reliably strong to make the conclusions that you have made? What about feedback effects? The authors have used this paradigm to test whether the PCs actually modulate the whisker movements but it is not clear whether a clear-cut conclusion about causative effects can be made. Perhaps it can, but the authors need to do a better job of fully explaining all the details of this experiment, including all the possible scenarios that might be occurring upon stimulation.

With the intensity and device that we used for optogenetic stimulation one obtains a stable elevated level of simple spike activity (Witter et al., 2013). The duration of 100 ms pulses of light was chosen in order to mimic the transient increase of simple spikes that were observed after 4 Hz air-puff stimulation. In fact, the middle-right part of Figure 5 D shows that the increase in simple spike activity is mainly in the first 100 ms after the puffs. Like the data of Proville et al., 2014, and those of Brown and Raman, 2018, our data allow for a rather clear-cut conclusion about the causal effects: the simple spikes in this cerebellar area can enhance the whisker movement. Even so, we agree that one cannot rule out secondary feedback mechanisms that may influence the outcome (Chaumont et al., 2013; Witter et al., 2013). We now describe this more clearly in the text.

5) The authors report that the 4Hz stimulation was sufficient to induce an increase in maximal protraction. The data are interesting, but what is the impact on the different circuits that control whisker behavior? It is not clear what global impact this manipulation has on the brain. I suppose I am having a bit of troubling linking this gross manipulation to the very specific spike data that was presented. This goes back to my previous query about anatomical pathways.

We agree with the reviewer that it is very hard to assert the impact of 4 Hz sensory stimulation on the different circuits that control whisker behavior. However, we based our research question on a series of previous studies from Egidio D'Angelo's lab on the sensitivity of the cerebellar cortex to stimulation in the theta band. This sensitivity comes from the property of resonance of the cerebellar granular layer (Ramakrishnan et al., 2016, Roggeri et al., 2008; D`Angelo et al., 2001). Because of that, inputs in theta range (4-8Hz) are more effectively transmitted and are able to induce plasticity more efficiently at several levels of the cerebellar cortex. Moreover, many reflexes are primarily encoded by spinal or brainstem circuits and modified by cerebellar activity (e.g. VOR adaptation and eyeblink conditioning). Bellavance and colleagues (2017) have recently shown that the whisker protraction reflex is brainstem-mediated and we know that the cerebellum projects to these areas. So, our data expand on these findings, analagous to other reflex adaptation mechanisms. We have now explained this better in the text, citing the appropriate references.

6) In the third paragraph of subsection “Tetanic 4 Hz stimulation leads to acceleration of the simple spike response and to stronger protraction of the whiskers”, the authors report that complex spike and simple spike responses to each air puff were weakened…. However, it is not so clear to me why this is the case. Was this result predicted? If yes (or no), what was the rationale?

The main motivation to include these responses was to describe what happened during the induction period. Inspired by this comment, we now also include the whisker responses during induction. In line with the Purkinje cell responses, they were diminished during 4 Hz (relative to 0.5 Hz) stimulation.

7) In subsection “Tetanic 4 Hz stimulation specifically potentiates simple spike firing in PCs with weak complex spike responses”, what is the authors definition of a "strong" complex spike responses? Are you talking about frequency, probability, amplitude, duration?

This has now been clarified in the new version; we mean response probability.

8) I do not understand why 3 out of the 13 L7-PP2B mice did show touch evoked whisker protractions. Being a genetic experiment, there must be some explanation for the presence of responses in these 3 animals.

We thank the reviewer for pointing this out. First of all, the amplitude of the touch-induced whisker protraction was not altered in the mutants (new Figure 8—figure supplement 2), indicating that the baseline reflex, which is brainstem-mediated (Bellavance et al., 2017), was still present. The reflex adaptation was impaired in both cerebellar mutant mouse lines, however. In the previous version, we showed a black-and-white approach (adaptation or no adaptation). We understand that this created confusion as it did not take the variation appropriately into account. We therefore replaced the pie diagrams with a more quantitative analysis, which shows that the observed variation in the mutants is well within the normal variation and does not indicate the presence of reflex adaptation.

[Editors’ note: the author responses to the re-review follow.]

We are pleased to consider acceptance of your paper for publication pending one correction. The statement "… whisker retraction is largely mediated by a passive process.…" is an overstatement. Retraction can be active – as pointed out in the manuscript "Rhythmic whisking in rat: Retraction as well as protraction is under active muscular control" by Berg et al., 2003, as well as in Hill et al. (J Neurosci 2008) and Bellavance et al., 2017. Retraction occurs through the action of the maxolabialis and nasolabialis extrinsic retractor muscles. These muscles are driven by input from the preBotzinger complex (Moore et al., 2013; Deschenes et al., 2017) as well as from other sources. Thus the retraction seen in Figure 1C,E can in principle be an active process. Please correct the text and remove the call-out "passive" from Figure 1.

We appreciate your comment on the active nature of retraction and modified the text and Figure 1 accordingly. In addition, we included the two references suggested by you to substantiate the possible involvement of active processes during whisker retraction.